# Analyses of a chromosome-scale genome assembly reveal the origin and evolution of cultivated chrysanthemum

Aiping Song[1,7], Jiangshuo Su[1,7], Haibin Wang[1,7], Zhongren Zhang [2,7], Xingtan Zhang [3,7], Yves Van de Peer[1,4,5], Fei Chen[6], Weimin Fang[1], Zhiyong Guan[1], Fei Zhang[1], Zhenxing Wang [1], Likai Wang[1], Baoqing Ding[1], Shuang Zhao[1], Lian Ding[1], Ye Liu[1], Lijie Zhou[1], Jun He[1], Diwen Jia[1], Jiali Zhang[1], Chuwen Chen[1], Zhongyu Yu[1], Daojin Sun[1], Jiafu Jiang [1]✉, Sumei Chen[1]✉ & Fadi Chen [1]✉

Chrysanthemum (*Chrysanthemum morifolium* Ramat.) is a globally important ornamental plant with great economic, cultural, and symbolic value. However, research on chrysanthemum is challenging due to its complex genetic background. Here, we report a near-complete assembly and annotation for *C. morifolium* comprising 27 pseudochromosomes (8.15 Gb; scaffold N50 of 303.69 Mb). Comparative and evolutionary analyses reveal a whole-genome triplication (WGT) event shared by *Chrysanthemum* species approximately 6 million years ago (Mya) and the possible lineage-specific polyploidization of *C. morifolium* approximately 3 Mya. Multilevel evidence suggests that *C. morifolium* is likely a segmental allopolyploid. Furthermore, a combination of genomics and transcriptomics approaches demonstrate the *C. morifolium* genome can be used to identify genes underlying key ornamental traits. Phylogenetic analysis of *CmCCD4a* traces the flower colour breeding history of cultivated chrysanthemum. Genomic resources generated from this study could help to accelerate chrysanthemum genetic improvement.

Cultivated chrysanthemum (*Chrysanthemum morifolium* Ramat., Chinese name "Ju Hua") is a perennial herbaceous ornamental plant that belongs to Asteraceae (also known as Compositae), the largest family of dicotyledons, with more than 1600 genera and 25,000 species[1]. In addition to being known for its outstanding beauty, cultivated chrysanthemum also holds a key evolutionary position in flowering plants as a representative of Asteraceae. Chrysanthemum originated in China and has been cultivated for over three thousand years[2,3]. From the 8th century AD to 17th century, several chrysanthemum cultivars were successively introduced to Japan and Europe[4], further expanding their breeding for ornamental purposes. At present, chrysanthemum is one of the most economically important floral plants worldwide[5], due to

[1]State Key Laboratory of Crop Genetics & Germplasm Enhancement and Utilization, Key Laboratory of Landscaping, Key Laboratory of Flower Biology and Germplasm Innovation (South), Ministry of Agriculture and Rural Affairs, Key Laboratory of Biology of Ornamental Plants in East China, National Forestry and Grassland Administration, College of Horticulture, Nanjing Agricultural University, Nanjing, Jiangsu 210095, China. [2]Novogene Bioinformatics Institute, Beijing 100083, China. [3]Shenzhen Branch, Guangdong Laboratory for Lingnan Modern Agriculture, Genome Analysis Laboratory of the Ministry of Agriculture, Agricultural Genomics Institute at Shenzhen, Chinese Academy of Agricultural Sciences, Shenzhen, Guangdong 518120, China. [4]Department of Plant Biotechnology and Bioinformatics, Ghent University, VIB Center for Plant Systems Biology, 9052 Ghent, Belgium. [5]Center for Microbial Ecology and Genomics, Department of Biochemistry, Genetics and Microbiology, University of Pretoria, Pretoria 0028, South Africa. [6]College of tropical crops, Sanya Nanfan Research Institute, Hainan University & Hainan Yazhou Bay Seed Laboratory, Sanya, Hainan 572025, China. [7]These authors contributed equally: Aiping Song, Jiangshuo Su, Haibin Wang, Zhongren Zhang, Xingtan Zhang. ✉e-mail: jiangjiafu@njau.edu.cn; chensm@njau.edu.cn; chenfd@njau.edu.cn

the striking diversity of its flower shapes and colours, and accounts for a large proportion of flower industry production, applying as cut-flower, garden and potted types. Additionally, chrysanthemum is widely used in the medical, food, and beverage industries because of its nutritional and biologically active components.

The *Chrysanthemum* genus harbours more than 41 species with varying ploidies, ranging from 2n = 2x to 10x, and natural hybridisation is prevalent in this genus[3,6]. Cultivated chrysanthemum is a self-incompatible complex hexaploid frequently showing a chromosome number of 54[7,8]. However, whether chrysanthemum should be genetically classified as an autopolyploid or allopolyploid has long been controversial. Abundant studies have suggested that several wild *Chrysanthemum* species might have contributed to current complex hybrid chrysanthemum cultivars[9–11]. Hence, following a long period of domestication and artificial selection, cultivated chrysanthemum now displays a much higher diversity of ornamental features than other species of the *Chrysanthemum* genus. However, the evolutionary history of the *Chrysanthemum* genus is poorly understood, and the actual ancestry of cultivated chrysanthemum remains elusive.

An available high-quality reference genome sequence is key to elucidating the species' origin and evolutionary history as well as the genetic basis underlying its phenotypic diversity, which can further accelerate future breeding processes[12]. Despite the economic importance and evolutionary significance of cultivated chrysanthemum, its genome has not yet been deciphered, mainly due to its polyploidy, high repetitiveness, high heterozygosity, and large size. Recently, two scaffold-level *Chrysanthemum* genomes, i.e., the ones of *C. nankingense*[13] and *C. seticuspe*[14], as well as three chromosome-scale genomes, i.e., the ones of *C. seticuspe*[15], *C. makinoi*[16], and *C. lavandulifolium*[17], have been reported. These valuable resources allow for dissecting the evolutionary history of the *Chrysanthemum* genus. However, these genomes all come from diploid wild species and therefore lack critical genetic information of chrysanthemum cultivars, making them less optimal for illuminating the complex genetic basis of phenotypic variation in hexaploid cultivated chrysanthemum. Several transcriptome studies have allowed us to preliminarily isolate a number of homologous genes involved in the regulation of ornamental traits in chrysanthemum[18–22], but it remains challenging to reveal novel determinants of these traits through forward genetics approaches due to the lack of a reference genome sequence for cultivated chrysanthemum.

Here, we present a chromosome-scale genome assembly of *C. morifolium* using a haploid line (n = 3x = 27) derived from pot variety 'Zhongshanzigui'[23]. We infer that the recent whole-genome triplication (WGT) and smaller-scale duplications contribute to the expansion of the known genes related to flower development in chrysanthemum and identify several candidate genes underlying petal shape. Furthermore, we trace the flower colour evolution history of cultivated chrysanthemum based on the phylogenetic analysis of *CCD4a* genes. The *C. morifolium* reference genome provides insights into the evolution of *Chrysanthemum* genus and is valuable for future genetic and molecular biology studies of chrysanthemum.

## Results

### Genome sequencing, assembly, and annotation

The genome size of the haploid line (Supplementary Fig. 1b, d) was estimated to be approximately 8.47-8.88 Gb by *K*-mer analysis using 1070.20 Gb clean short reads (Supplementary Fig. 2a–d and Supplementary Tables 1 and 2), which was slightly smaller than the size estimated by flow cytometry (9.02 Gb; Supplementary Fig. 2e). The genome survey also revealed a high repeat content (72.48% ~ 88.85%) in *C. morifolium* (Supplementary Table 1). We initially corrected and assembled the 1022.3 Gb (120.70×) PacBio continuous long reads (CLR) into contig sequences using Falcon and polished them using Quiver. The consensus sequences were then scaffolded by integration

with 907.5 Gb (107.20×) of genome sequences derived from a 10X Genomics library and polished using short reads. Next, we used the AllHiC algorithm[24] to improve the *C. morifolium* genome assembly based on 1002.9 Gb (118.50×) of Hi-C data (Supplementary Table 2 and Supplementary Note 1). This allowed 96.46% of the original assembly sequences to be anchored to 27 pseudochromosomes with lengths ranging from 214.51 to 343.67 Mb (Fig. 1a, Table 1 and Supplementary Tables 3 and 4). The contig and scaffold N50 sizes of the final chromosome-level genome assembly of *C. morifolium* were 1.87 Mb and 303.69 Mb, respectively (Table 1). The Hi-C interaction heatmap clearly demonstrated that the 27 pseudochromosomes could be clustered into nine homoeologous groups (Fig. 1b). The three pseudochromosomes within a homoeologous group were highly similar in terms of their size, gene number and repeat element content (Supplementary Table 4). Plots of the synteny relationships and *Ka/Ks* ratios of each syntenic gene pair clearly showed a high degree of conserved synteny, with no substantial overall *Ka/Ks* ratio difference between any two chromosomes (Fig. 1a).

To evaluate the assembled genome quality, we aligned 105,996 expressed sequence tags (ESTs) of *C. morifolium* downloaded from the US National Center for Biotechnology Information (NCBI) database to the assembled genome and found that 96.43% of the ESTs could be mapped (Supplementary Table 5). BUSCO and Core Eukaryotic Genes Mapping Approach (CEGMA) analyses found that 97.70% and 98.39% of the total eukaryotic conserved genes, respectively, were identified in the chromosome-scale genome of *C. morifolium* (Supplementary Table 6), which is comparable to results for the *C. seticuspe* genome[15] but higher than the percentages reported for the other released genomes of Asteraceae species (Supplementary Data 1). The annotation of long terminal repeats (LTRs) revealed an LTR Assembly Index (LAI)[25] score of 27.99, meeting the gold standard for reference genomes, comparable to the scores of autotetraploid alfalfa (22.30)[26] and allohexaploid *Echinochloa colona* (22.5)[27]. The high fidelity of the final assembly was also supported by the high mapping rate of 98.81% and the estimated high base accuracy of 57 homozygous single nucleotide polymorphisms (SNPs) per 1 Mb (0.0057%) by using Illumina sequencing data (Supplementary Table 7). The top 10× (approximately 80 Gb) longest PacBio long reads, ranging from 36.32 to 156.36 kb, were extracted and mapped against the assembled genome, and a large majority (82.24%) of these reads could be uniquely mapped to only one chromosome with more than 80% alignment length, indicating that most of the chromosomes were phased correctly. Furthermore, 82.15% of the transcriptomes could be mapped back to the final assembly (Supplementary Table 8), further supporting a high level of genome integrity. Taken together, the above results indicate a high degree of continuity and completeness of the *C. morifolium* genome.

Additionally, a chromosome-level assembly of diploid *C. nankingense* (2n = 2x = 18), a potential progenitor of cultivated chrysanthemum, was generated for comparative studies using a similar hybrid assembly strategy (Supplementary Table 9 and Supplementary Note 2). The genome assembly anchored 2.87 Gb of the contigs to 9 pseudochromosomes, representing 93.16% of the estimated 3.09 Gb *C. nankingense* genome (Supplementary Table 10). The high contig N50 length of 5.98 Mb and scaffold N50 length of 353.78 Mb indicate a superior genome assembly compared to the previously published scaffold-level *C. nankingense* genome[13]. Intergenomic comparison analysis revealed, as expected, a distinct 3-to-1 syntenic relationship between *C. morifolium* and *C. nankingense*. Only a few inversion or translocation regions were identified (Fig. 1c).

Based on de novo and homology-based predictions and transcriptome data (Supplementary Note 3), we predicted a total of 138,749 protein-coding genes in the *C. morifolium* genome, which is considerably greater than the number annotated for other Asteraceae plants, ranging from 28,310 genes for globe artichoke to 74,259 for *C. seticuspe*. The average lengths of total gene regions, coding

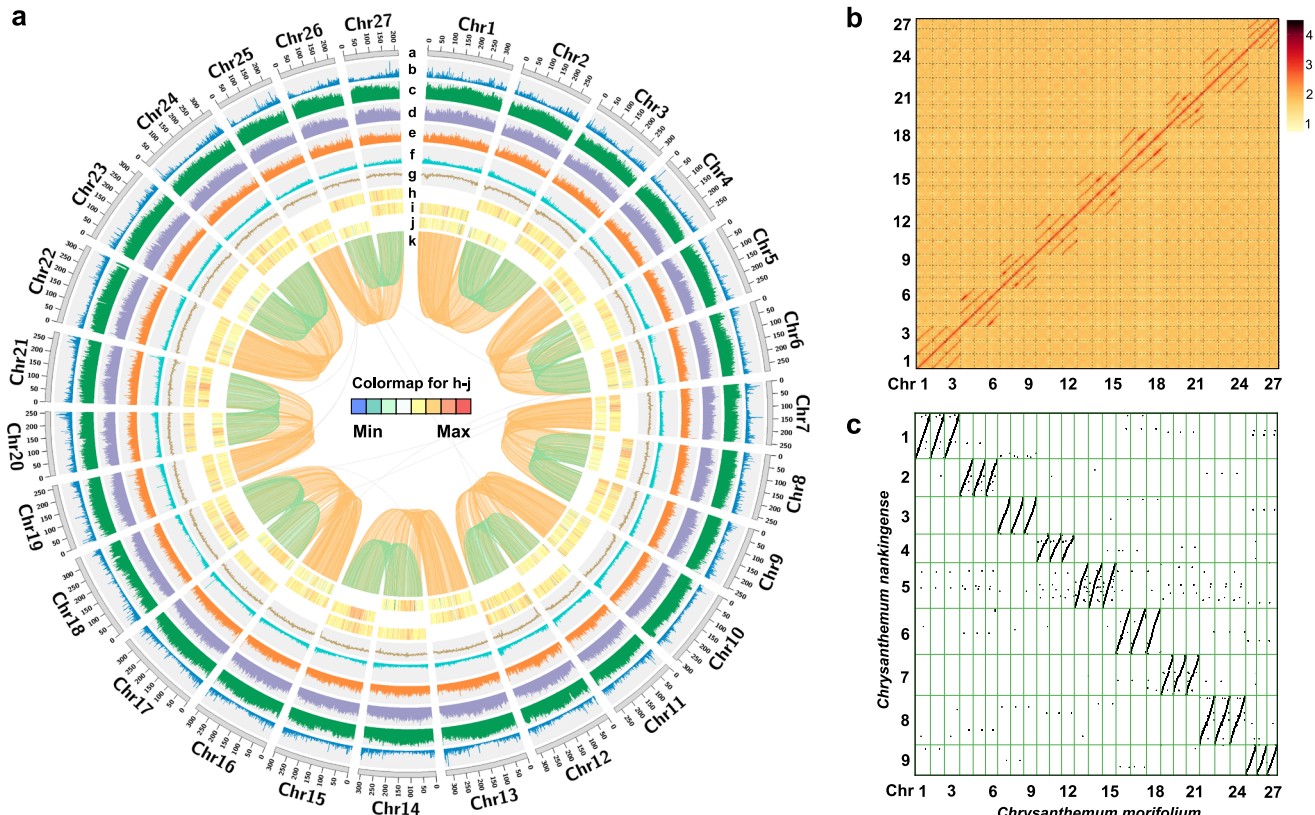

**Fig. 1 | Overview of the *C. morifolium* genome. a** Circus plot showing the genomic features of the haploid *C. morifolium* cv. 'Zhongshanzigui'. Circular tracks from outside to inside indicate: **a** the pseudomolecules; **b** gene density; **c** density of LTR transposons; **d** density of *Copia* transposons; **e** density of *Gypsy* transposons; **f** density of DNA transposons; **g** GC content; **h** *Ka/Ks* of syntenic gene pairs identified between traid1 and either traid2 or traid3 within a homoeologous chromosome group; **i** *Ka/Ks* of syntenic gene pairs identified between traid2 and either traid1 or traid3 within a homoeologous chromosome group; **j** *Ka/Ks* of syntenic gene pairs identified between traid3 and either traid1 or traid2 within a homoeologous chromosome group; **k** syntenic genes between different pseudochromosomes. Orange lines indicate syntenic blocks between traid1 and traid2 and traid3, green lines indicate syntenic blocks between traid2 and traid3. **b** Hi-C heatmap showing the chromosomal interactions. Each homoeologous group contains three pseudomolecules, there are few linkages between different homoeologous groups, indicating high quality chromosome-level scaffolding. **c** Genome comparison between *C. morifolium* (Chr1-27) and diploid *C. nankingense* (Cna1-9). Source data are provided as a Source Data file.

sequences (CDSs), exon sequences, and intron sequences were 3912, 1090, 242 and 806 bp, respectively (Supplementary Data 2). On average, each predicted gene contained 4.50 exons, and the gene density throughout the genome was approximately one gene per 58.74 kb, with 134,450 genes (96.90%) present on chromosomally anchored contigs. Genes were distributed unevenly, being more abundant towards the ends of the chromosomal arms as expected (Fig. 1a). Approximately 99.3% of the predicted protein-coding genes could be functionally annotated via searches in the NR (90.97%), Swiss-Prot (75.58%), GO (90.78%), KEGG (70.28%), InterPro (98.75%) and Pfam (72.60%) databases (Supplementary Table 11). In addition, we identified 50,848 noncoding RNA (ncRNA) genes encoding 2868 ribosomal RNAs (rRNAs), 4102 transfer RNAs (tRNAs), 2280 microRNAs (miRNAs), and 41,598 small nuclear RNAs (snRNAs) (Supplementary Table 12). Using RNA-seq data from flower buds and nine organs, including roots, stems, shoot leaves, and flower organs dissected from ray florets and disc florets, 103,287 (74.44%) of the identified genes were expressed in at least one tissue, and 57,869 (41.71%) were expressed in all organs analysed (Supplementary Table 13).

## Comparative genomics and evolutionary analysis
We inferred the phylogenetic position and divergence times among cultivated chrysanthemum and 14 other plant species (Supplementary Note 4), including 7 asterids I (*C. nankingense, C. seticuspe*, sunflower, lettuce, globe artichoke, *Artemisia annua* and carrot), 2 asterids II

(tomato and coffee), 1 rosids (grape), 1 Ranunculales (columbine), 2 monocots (rice and maize) and 1 basal angiosperm (Amborella). We constructed a consensus set containing 94,552 genes to represent the monoploid genome of *C. morifolium*. A total of 507,449 genes were clustered into 48,644 orthologous gene families (orthogroups), including 6193 gene families shared by all 15 species (Supplementary Fig. 3 and Supplementary Data 3). By comparing seven Asteraceae species, we identified 10,234 gene families shared by all members and 3543 families containing 9638 genes that appeared unique to *C. morifolium* (Supplementary Fig. 4). In addition, gene family evolution analysis revealed 1684 gene families showing possible expansion in *C. morifolium*, whereas 1926 gene families showed contraction (Fig. 2a). Functional enrichment analysis demonstrated that these expanded gene families were mainly involved in terpene biosynthetic processes, auxin metabolic processes, regulation of hormone levels, regulation of anthocyanin biosynthetic processes, and regulation of double-strand break repair via homologous recombination (Supplementary Fig. 5 and Supplementary Data 4).

As expected, a phylogenetic tree (Fig. 2a) constructed using 491 single-copy gene families agreed with previous studies[13,28–30]. Molecular dating suggested that Asteraceae diverged from carrot (Apiaceae) at approximately 92.4 Mya. Within Anthemideae, cultivated chrysanthemum diverged from the most recent common ancestor of *C. nankingense* and *C. seticuspe* ~3.7 Mya, following the divergence of *Artemisia annua* approximately 6.5 Mya. Anthemideae species are

**Table 1 | The statistics for genome assembly and annotation of *C. morifolium***

|  | Number | Size |
|---|---|---|
| **Genome assembly** | | |
| GC content | - | 36.18% |
| Total contigs | 19,524 | 8125.34 Mb |
| Contig L50/N50 | 1311 | 1.87 Mb |
| Contig L90/N90 | 4714 | 387.91 Kb |
| Total scaffolds | 5953 | 8154.32 Mb |
| Scaffold L50/N50 | 13 | 303.68 Mb |
| Scaffold L90/N90 | 25 | 232.21 Mb |
| Pseudochromosomes | 27 | 7865.51 Mb |
| BUSCO completeness of assembly (%) | - | 97.71 |
| **Genome annotation** | | |
| Repetitive sequences | 83.92% | 6843.49 Mb |
| Protein-coding genes | 138,749 | 596.19 Mb |
| Genes in pseudochromosomes | 134,450 | 581.75 Mb |
| BUSCO completeness of annotation (%) | - | 97.89 |

more related to sunflower (Heliantheae, diverged ~34.8 Mya) than to lettuce and globe artichoke (Cichorieae, diverged ~39.3 Mya).

It is widely recognised that whole-genome duplication (WGD) or WGT events have a profound effect on shaping plant genome evolution and speciation[31]. We first estimated the synonymous substitution rate ($Ks$) values of collinear paralogous gene pairs within monoploid *C. morifolium*, *C. nankingense* and *C. seticuspe*, sunflower, and globe artichoke to identify WGD/WGT events (Fig. 2b). These results, together with previous reports[13,29,32], demonstrated that *C. morifolium* has a complex palaeopolyploid history. In addition to the ancient WGT-γ (approximately 122-164 Mya) shared by all core eudicots, WGT-1 (~57 Mya) shared by asterids II, and WGD-2 (~38 Mya) in sunflower, we identified a recent lineage-specific WGD event shared by all three *Chrysanthemum* species that occurred approximately 6 Mya, showing signature $Ks$ peaks at approximately 0.1, in agreement with a previous study[13]. Moreover, genomic dot plot analysis clearly illustrated a 3:1 syntenic relationship within *C. morifolium* monoploid pseudochromosomes and a 3:1 syntenic relationship between *C. morifolium* and globe artichoke (Supplementary Fig. 6a, b), which experienced only the Asteraceae-shared WGT-1 event[33]. Similar relationships were detected within *C. nankingense* and between *C. nankingense* and globe artichoke (Supplementary Fig. 6c, d). These results provide additional evidence that the inferred recent polyploidization event in the common ancestor of *Chrysanthemum* species was more likely to be a WGT event than a WGD event (herein termed WGT-2). We also calculated the $Ks$ values between paralogous genes within each of the nine homoeologous groups (Fig. 2b) and found another signature peak at approximately 0.05 (~3 Mya) (Fig. 2b), suggesting that apart from the lineage-specific WGT-2 event shared by all three *Chrysanthemum* species, *C. morifolium* underwent a very recent polyploidization event. This may be one potential reason for the observed highly conserved triplicate structure without dramatic chromosomal rearrangements in the *C. morifolium* genome (Fig. 1c).

Gene duplication has been considered a major force for evolution[34]. In the *C. morifolium* genome, 136,137 duplicated genes were divided into five categories, i.e., whole-genome duplicates (WGD, 61.5%), dispersed duplicates (DSD, 43.1%), transposed duplicates (TRD, 17.0%), proximal duplicates (PD, 6.9%) and tandem duplicates (TD) (Supplementary Fig. 7 and Supplementary Note 4). Higher $Ka/Ks$ ratios were found for TRD and PD gene pairs (Supplementary Fig. 8a), suggesting an ongoing, continuous process of transposed and proximal duplications and more relaxed selective pressure for duplicates generated via these two modes, while duplications resulting from all five modes showed similar $Ks$ values (Supplementary Fig. 8b). Notably, duplications via all five modes were associated with divergent enriched GO terms. For example, 'WGD genes' showed significant enrichment of GO terms involved in various developmental processes, regulation of transcription and signal transduction; the PD gene-enriched categories were implicated in recognition of pollen and sesquiterpene biosynthetic processes, while the TD genes were assigned to monoterpenoid biosynthetic and terpenoid metabolic processes as well as stress responses (Supplementary Fig. 9). Terpene synthases (TPSs) are the key enzymes responsible for the biosynthesis of terpenoid compounds in green plants. We found that the genomes of Anthemideae species contained many more TPSs than those of other Asteraceae plants, specifically for the TPS-a and TPS-b subfamilies (Supplementary Figs. 10 and 11), which are predominantly involved in the synthesis of sesquiterpenes and monoterpenes, respectively[35]. Several tandem-duplicated TPS-a and TPS-b genes were identified. In particular, 13 TPS-a copies and 10 TPS-b copies were densely distributed on chromosomes 27 and 19, respectively (Supplementary Fig. 10).

Repetitive sequences were identified using a combination of ab initio and homology-based approaches. In total, 83.38% of the assembled sequences were annotated as repetitive elements (TEs), including 79.59% retrotransposons, 8.63% DNA transposons and 0.92% simple repeats and satellites (Supplementary Table 14). The proportion of TEs identified in *C. morifolium* is, to our knowledge, the highest among the sequenced Asteraceae species, ranging from 58.4% in globe artichoke to 74.7% in sunflower. LTR retrotransposons were found to constitute 72.96% of the *C. morifolium* genome, with the *Copia* (40.40%) and *Gypsy* (24.79%) subfamilies accounting for the largest percentages. The *Copia* element density increased from the ends of chromosomes towards their centres, yet there was a relatively even distribution of *Gypsy* elements along the chromosomes (Fig. 1a). Notably, there were large differences in *Copia* and *Gypsy* proportions in Asteraceae. In contrast to the situation of more abundant *Copia* elements in the *C. morifolium* (1.63), *C. nankingense* (1.74), *C. seticuspe* (1.28)[15], *C. makinoi* (3.66)[16] and *C. lavandulifolium* (1.3)[17] genomes, the *A. annua* (0.86)[28], sunflower (0.26)[29], and newly released *Mikania micrantha* (0.37)[32] genomes harboured more *Gypsy* elements (Fig. 2c). These findings suggest that the amplification of LTR/*Copia* elements was a major driving force for the inflation of the *Chrysanthemum* genomes. The mechanisms of these distinct behaviours of *Copia* and *Gypsy* elements remain to be elucidated. We found very recent bursts (<0.5 Mya) of LTRs in *C. morifolium*, *C. nankingense* and sunflower after the three species diverged, which was slightly later than the LTR bursts identified in *A. annua* (~ 1 Mya) and globe artichoke (~ 2 Mya). Moreover, the bursts of *Copia* and *Gypsy* retrotransposon insertions in *C. morifolium* occurred close in time, and there was no predominant difference in predicted time of *Copia*/*Gypsy* bursts among chromosomes (Supplementary Fig. 12). Collectively, these results suggest that the *C. morifolium* genome is characterised by active transposition, and the accumulation of TE insertions is the main reason for the expansion of the cultivated chrysanthemum genome.

## Origin of cultivated chrysanthemum

Although the origin of cultivated chrysanthemum has attracted considerable attention, the wild progenitors that contributed to cultivated chrysanthemum remain uncertain. Previous phylogenetic studies aimed at identifying these progenitor species, often based on traditional morphological and cytological taxonomy or a limited number of molecular markers, have produced inconsistent results[3]. By taking advantage of the *C. morifolium* genome, we resequenced 12 Chinese wild *Chrysanthemum* species that are assumed to be the most likely ancestral species of cultivated chrysanthemum, with an average coverage depth of approximately 8.5× for each accession (Supplementary Fig. 13, Supplementary Data 5). The phylogenetic tree (Fig. 3a) generated with 11,755 SNPs clearly illustrated that *C. rhombifolium*, which

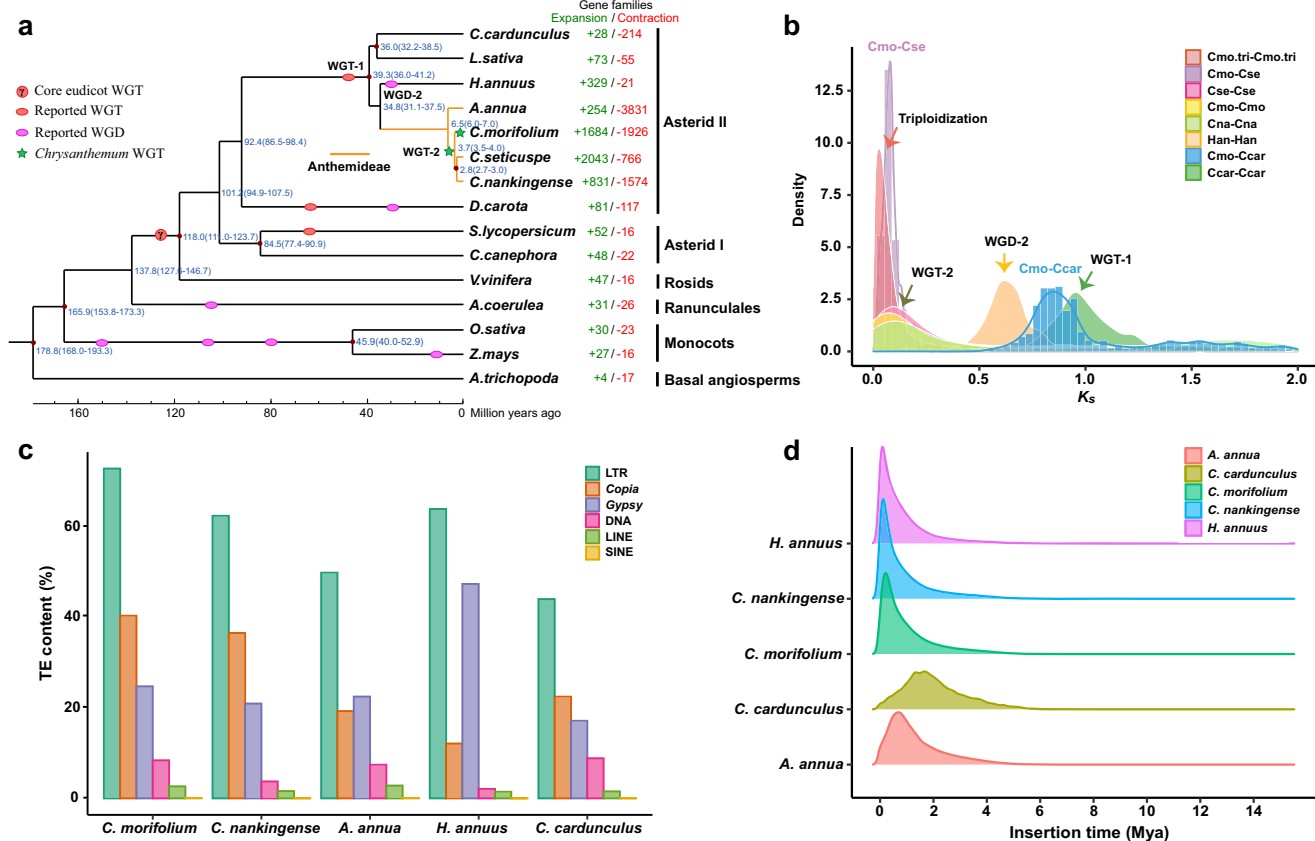

**Fig. 2 | Genome evolution of *C. morifolium*. a** Phylogenetic tree showing the evolutionary relationship of cultivated chrysanthemum and 14 other flowing plants. Expansion (green) and contraction (red) of gene family numbers are shown on the right. The red circles on some nodes represent fossil calibrations points, obtaining from the TIMETREE website (http://www.timetree.org/)[112]. Predicted divergence times (Mya, million years ago) are labelled in blue at other intermediate nodes as estimated by Maximum Likelihood (PAML), and the numbers in parentheses are

95% confidence intervals of the divergence time between different clades. The WGD and WGT events are marked with coloured dots. **b** *Ks* distribution of syntenic paralogues found within *C. morifolium* and five other Asteraceae sequenced species. The polyploidization events are referenced on the peaks. **c** Comparison of repetitive sequences among *C. morifolium* and *C. nankingense, A. annua, H. annuus, C. cardunculus.* **d** Insertion time distributions of intact LTRs in different Asteraceae species. Source data are provided as a Source Data file.

bears white ray florets, and tetraploid *C. indicum* from Nanjing, might be more closely related to cultivated chrysanthemum than other species, consistent with the findings that *C. rhombifolium* and *C. indicum* (Nanjing) shared the largest numbers of fragments specific with *C. morifolium* (Fig. 3b). In addition, identity score (IS) analysis clearly illustrated that *C. indicum* (Nanjing) (0.866), *C. rhombifolium* (0.853), *C. indicum* (Tianzhushan) (0.847), *C. indicum* (Hubei) (0.844), *C. dichrum* (0.841) and *C. potentilloides* (0.833), which formed an independent clade, also exhibited higher genomic similarity to cultivated chrysanthemum than the other wild *Chrysanthemum* species (Fig. 3c and Supplementary Data 6). Notably, the *C. indicum* accessions with different geographic regions and ploidy levels were grouped into three main clades and exhibited even more genetic diversity than was observed between species (Fig. 3a), which has also been found in other studies[3]. We speculate that interspecific hybridisation within the *Chrysanthemum* genus as well as *C. indicum* habitat diversity might have contributed to this genetic variation.

In a further attempt to track introgression events, we calculated the average IS values within each 100 kb sliding window across the *C. morifolium* genome. As a result, massive small hotspots with higher IS values and a few large hotspots derived from most of these wild *Chrysanthemum* species were observed in cultivated chrysanthemum (Supplementary Fig. 14), suggesting that extensive and multiple genomic introgression might have contributed to the formation of cultivated chrysanthemum as well as the complex reticulate evolution history of *Chrysanthemum* species.

*C. nankingense* has long been suspected to be a diploid progenitor of cultivated chrysanthemum[13]. The two genomes assembled in this study enable us to study gene loss and retention following triploidization of *C. morifolium*. Approximately 74.44% of all genes in *C. nankingense* were syntenic with at least one homologous copy in *C. morifolium*, while 55.10% of genes had three homologous copies retained in *C. morifolium* (Supplementary Fig. 15). However, we could not identify a dominant chromosome set in the *C. morifolium* genome that could be clearly attributed to *C. nankingense*, unlike the situation in allopolyploid strawberry[36] and broomcorn millet[37], suggesting that *C. nankingense* might not be a direct ancestral donor of cultivated chrysanthemum. This finding was supported by the SNP-based phylogenetic tree (Fig. 3a) and IS analyses (Fig. 3c). Taking advantage of the assembly of *C. seticuspe* genome[15], we performed a similar gene retention analysis and found that *C. seticuspe* was less closely related to *C. morifolium* than *C. nankingense* (Supplementary Fig. 16 and Supplementary Data 7).

It remains unclear whether cultivated chrysanthemum has an autopolyploid or allopolyploid origin. Distinguishing between different forms of polyploidy is not always straightforward. Generally, chromosome behaviour, fertility, segregation ratios, and morphology, coupled with genetic data, are principal criteria for distinguishing between autopolyploids and allopolyploids[38]. Previous cytological studies revealed that hexaploid cultivated chrysanthemum shows diploid-like meiosis in which bivalents dominate, but univalents and multivalents (mainly quadrivalents) are rarely observed[39–41]. However,

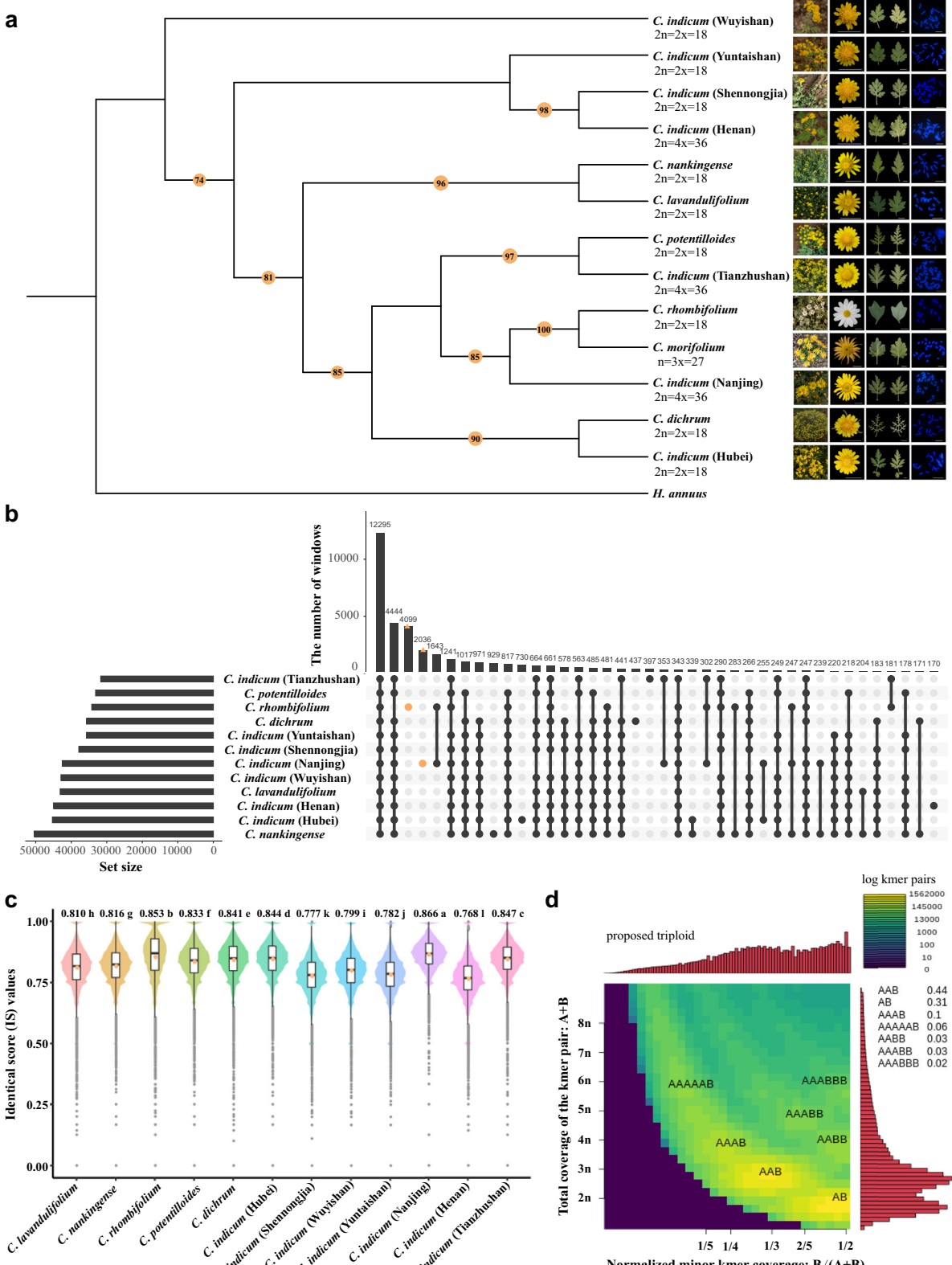

the disomic or polysomic inheritance of chrysanthemum represents extreme cases within a continuum[42,43]. Our karyotype analysis conducted during pollen mother cell (PMC) meiosis showed that 18 chromosomes formed nine bivalents, while nine chromosomes remained univalent (9II + 9I) in the sequenced haploid, and the colchicine-induced doubled haploid behaved like an allopolyploid and almost exclusively formed bivalents (27 II) (Supplementary Fig. 1f and Supplementary Note 5). Phylogenetic analysis of each of the

homoeologous groups based on the unique SNPs showed that the three homoeologous chromosomes within the same group were all separated from each other except for Chr17 and Chr18 (Supplementary Fig. 17), which is consistent with the possibility of an allopolyploid origin for *C. morifolium*.

Nevertheless, we found a pattern of uniform coverage of different wild species in comparison with the *C. morifolium* genome (Supplementary Fig. 13), which differs from what has been observed for

**Fig. 3 | The origin of the cultivated chrysanthemum. a** Maximum-likelihood phylogenetic tree of *Chrysanthemum* plants using sunflower (*H. annuus*) as the outgroup. Numbers above the branches indicate the bootstrap values. The morphology and cytological characterisations of *C. morifolium* cv. 'Zhongshanzigui' and 12 re-sequenced diploid or tetraploid wild *Chrysanthemum* species are shown on the right. **b** UpSet plot showing the number of 100 kb non-overlapping sliding windows with coverage depth larger than 4×. **c** The identical score (IS) values distribution of 12 wild *Chrysanthemum* species. The higher IS values indicate a closer relationship with *C. morifolium*. The central line for each box plot is the median, the top and bottom edges correspond to the first and third quartiles, and the whiskers represent the 1.5 inter-quartile range extending from the edges of the box. The grey points are outliers. The average IS value for each species is indicated as orange dots and marked on the top of box plot. Data were analyzed using one-way ANOVA followed by a two-tailed Tukey's honestly significant difference (HSD) multiple comparison test ($n = 70,524$), and different lowercase letters indicate a statistically significant difference at $P < 0.01$. **d** Smudgeplots analysis based on kmers for *C. morifolium*. Source data are provided as a Source Data file.

allotetraploid tobacco[44] and cotton[45]. This incongruence might occur because the species within the *Chrysanthemum* genus are closely related; the extant diploid or tetraploid wild *Chrysanthemum* species probably have also undergone extensive reticulate evolution with high heterozygosity; and the putative primitive ancestor of cultivated chrysanthemum may have gone extinct, as described by Ma et al.[46]. To verify that the results were not due to the selected potential progenitors, we also used a 13-mers clustering approach (Supplementary Note 5) based on the distribution of TEs, which did not require access to or even knowledge of living representatives of the progenitor lineages[47]. Although each set of the three homoeologous chromosomes was clearly clustered based on the identified 4719 chromosome-specific 13-mers, we found obvious differences in 13-mer counts among all homoeologous chromosomes except for Chr7-Chr8-Chr9 (Supplementary Fig. 18), suggesting a non-strict autopolyploid origin of *C. morifolium*. Moreover, the results indicated that the 17-mers with a depth >120x accounted for a larger portion (Supplementary Fig. 2a) compared with autopolyploid alfalfa[48]; a significant difference with respect to the *K*s distributions for syntenic orthologs between *C. morifolium* and *C. nankingense* was observed among four out of the nine homoeologous groups (Supplementary Fig. 19); slightly different gene retention patterns within each homoeologous group (Supplementary Figs. 15 and 16) further supported the possibility that *C. morifolium* might not be a strict autopolyploid. Intriguingly, a Smudgeplot analysis[49] implied that the genome structure of *C. morifolium* might be "AAB" or "AB" (Fig. 3d). Comparison between the homoeologous chromosomes in a triad by fluorescence in situ hybridisation (FISH) assays, also showed two homoeologous chromosomes often to be generally more similar compared to a third one but also presented differences (Supplementary Fig. 20 and Supplementary Data 8), supporting an "AA'B" relationship of homoeologous chromosomes in *C. morifolium*.

According to these findings together with evidence from previous reports[3,6,46,50,51] as well as comparisons with other typical allopolyploids and autopolyploids, we conclude that *C. morifolium* cv. 'Zhongshanzigui' is likely to be a specific segmental allopolyploid.

## Homoeolog expression bias in *C. morifolium*

To explore the transcriptional behaviour of *C. morifolium*, we compared genome-wide transcription levels of the annotated genes in different organs (Supplementary Note 6). Similar expression patterns were found for each homoeologous group across organs, except that the expression levels of genes located in the middle of chromosome 13 were significantly lower than those on chromosomes 14 and 15 (Supplementary Figs. 21 and 22). We focused on 11,438 genes showing a 1:1:1 correspondence in syntenic blocks across the three homoeologous chromosomes, referred to as triads. The global analysis combining data across all nine organs showed that 21.65% of syntenic triads presented no statistically significant differences in expression, being assigned to the balanced category, ranging from 19.20% in the Chr1-Chr2-Chr3 chromosome set to 23.50% in the Chr22-Chr23-Chr24 chromosome set (Supplementary Fig. 23); the proportion of syntenic triads with single-homoeolog dominance (41.73%) was significantly higher than that with single-homoeolog suppression (36.61%) (Supplementary Fig. 24), among which Chr8 dominance accounted for the largest proportion (24.7%). GO analysis indicated that the balanced triads were enriched in different biological processes among the nine homoeologous groups associated with various life processes. In particular, 'protein stabilisation', 'reproductive structure development', 'regulation of multicellular organismal development', 'regulation of transferase activity', 'leaf development', 'shoot system development', etc., were enriched in two homoeologous groups, while auxin transport (GO:0060918) was found in three homoeologous groups (Supplementary Data 9). Furthermore, we found a weak correlation between the synteny gene orientation and differential gene expression (Supplementary Note 6, Supplementary Tables 15 and 16), suggesting that the homoeolog expression bias in *C. morifolium* might not be caused by the change in gene orientation. These observations potentially represent the first steps towards neo- or sub-functionalization of *C. morifolium* genes.

## Genetic basis of flower evolution and development

Flower shapes have been under intensive selection during cultivated chrysanthemum improvement and continue to be a primary target of breeding programs. Cultivated chrysanthemum has a more diverse capitulum morphology than other Asteraceae species, which is largely determined by the colourful and multiform ray floret petals. According to the degree of corolla tube merging, florets can be divided into flat, spoon-like, and tubular ray florets (Supplementary Fig. 25c), but the genetic basis of floret diversity is not yet clear. Previous studies have shown that the identity of floral organs is regulated by MADS-box genes, as part of the well-characterised ABCE model[52]. Here, we identified all MADS-box (Supplementary Fig. 26 and Supplementary Data 10) and ABCE genes (Supplementary Fig. 25a) in the *C. morifolium* genome. The results showed that, as expected from its polyploidy nature, the *C. morifolium* genome possessed more MADS-box genes than other Asteraceae plants and *A. thaliana*, especially for the SEP and SVP clades. Synteny analysis, *K*s analyses of the orthologous gene pairs and duplication analyses revealed that both the recent WGT-2 as well as smaller-scale duplications contributed to the expansion of the MADS-box family members (Supplementary Fig. 6a, b), resulting in at least three very similar copies of each ABCE model gene in the *C. morifolium* genome. Compared with unduplicated orthologues, duplicates can undergo relaxed functional constraint, allowing them to diverge in both their gene sequence and expression patterns[53] and to undergo subfunctionalization or neofunctionalization. As shown in Supplementary Fig. 25a, the *PISTILLATA* (*PI*) paralogues indeed showed different expression patterns. One copy (evm.model.scaffold_4253.83) presented a higher expression level in vegetative organs, while the other members were highly expressed in the reproductive organs, implying that their functions might have undergone subfunctionalization. In contrast, the expression levels of the six paralogues of *AGAMOUS* (*AG*) in the disc florets of 9 varieties were significantly higher than those of the corresponding genes in the ray florets, suggesting the functional conservation of *AG* in determining flower organ identity. Gene expression analysis also indicated that the MADS-box B class genes *PI* and *DEFICIENS* (*DEF*)/*APETALA3* (*AP3*) might play important roles in petal and stamen development.

TCP family *CYCLOIDEA2* (*CYC2*) clade genes regulate floral symmetry, inflorescence architecture and reproductive organ

development[54,55]. Previous studies demonstrated that *CYC2* clade genes regulate the differentiation of ray and disc florets in Asteraceae species, such as sunflower, groundsel (*Senecio vulgaris*) and gerbera[54,56–58]. We identified 25 *CYC2-like* genes in *C. morifolium* (Supplementary Fig. 25b), which is a much greater number than the seven and eight members found in *C. nankingense* and sunflower, respectively. Phylogenetic analysis showed that the *CYC2a-like* genes of chrysanthemum have undergone several duplication events; genes from one subclade were highly expressed in reproductive organs, while genes from another subclade were highly expressed in vegetative organs, in accordance with the other *CYC2* clade genes (Supplementary Fig. 25b). Previous studies showed that the constitutive expression of *CmCYC2c* in *C. lavandulifolium* results in increases in both the number of ray florets and the length of the ray floret petal ligule[18,59]. However, there was no correlation between their expression and the degree of petal fusion, suggesting that petal fusion is controlled not only by *CYC2s* but also by other genes in *C. morifolium*.

To further determine the molecular genetic basis of petal shape variation, we performed a whole-genome bulked segregate analysis sequencing (BSA-seq) protocol in an F$_1$ population derived from a flat-type cultivar and a tubular-type cultivar (Fig. 4a, Supplementary Fig. 27 and Supplementary Note 7). Seventy-two consistent genomic regions associated with the degree of petal fusion were detected using ΔSNP-index and ED algorithms (Supplementary Data 11). In particular, the MYB-class gene *RAD6* (evm.model.scaffold_2712.53), acting downstream of *CYC/DICHOTOMA*, antagonistically to *DIVARICATA* (*DIV*) in the control of floral symmetry[60,61], was found in *qPT5-5* (Fig. 4b). The QTL was consistently detected across BSA-seq methods, located in the interval of 201.00-202.10 Mb on chromosome 5, with ΔSNP index and ED peaks of 0.91 and 1.65, respectively, for SNPs and 0.76 and 1.15 for InDels (Supplementary Fig. 28). *CmRAD6* was highly expressed in flat-type cultivars, and it was also a hub gene in the turquoise module that negatively correlated with the degree of petal fusion ($r = -0.92$; $P = 5.0E-04$) (Fig. 4c, d and Supplementary Fig. 29). Additionally, several candidates for flower development, such as *CYC2s*, *AGL6*, *MADS33* (*AGL12*), *IBL1*, *PI*, were identified via BSA-seq and weighted correlation network analysis (WGCNA) (Fig. 4d and Supplementary Fig. 28). Several auxin-responsive genes were also identified (Supplementary Data 12), suggesting that auxin might play a crucial role in petal development, consistent with our previous findings[21]. These results provide insights into the genetic architecture of petal shape in cultivated chrysanthemum, and the reference genome assembled herein offers major resources for accelerating future QTL fine mapping and breeding programs.

## Phylogenetic analysis of *CmCCD4a* revealed the flower colour breeding history of chrysanthemum

The flower colour diversity of cultivated chrysanthemum is mainly due to its colourful ray floret petals, which is one of the main ornamental traits of these plants. According to literature dating from the Tang Dynasty (618-907), cultivated chrysanthemum had only yellow ray floret petals, while various colours of ray floret petals in cultivated chrysanthemum were recorded in literature from the Song Dynasty (960-1279)[62]. One important question is how the petal colour of cultivated chrysanthemum changed from only yellow to a variety of flower colours. It has been shown that carotenoid cleavage dioxygenase (*CmCCD4a*) regulates the degradation of yellow pigments (carotenoids) in the petals of chrysanthemum ray florets[63] (Supplementary Note 8 and Supplementary Fig. 30c, e). We found that *CCD4a* was missing from the *C. morifolium* genome. Furthermore, the original chrysanthemum cultivar 'Zhongshanzigui' has purple ray floret petals (Supplementary Fig. 1a). However, the sequenced haploid material showed yellow or slightly orange-ray floret petals (Fig. 3a), and the flower colour could not be recovered by artificial whole-genome

doubling (Supplementary Fig. 1b, c). Therefore, we hypothesise that the *CCD4a* gene of wild *Chrysanthemum* species was introduced into cultivated chrysanthemum through artificial hybridisation but in low copy numbers, making it easy to lose after hybridisation, resulting in the return to yellow ray floret petals.

To verify this hypothesis and determine how *CCD4a* was introduced into cultivated chrysanthemum, we sought to amplify the *CCD4a* genes of 40 cultivated chrysanthemum varieties and 14 related wild species of *Chrysanthemum* genera. The results showed that there is no *CCD4a* gene in the genomes of cultivated varieties and wild relatives with yellow or orange ray floret petals. Finally, we cloned 78 *CCD4a* sequences from 23 cultivated chrysanthemum varieties and nine wild relatives of *Chrysanthemum* genera. Phylogenetic analysis showed that *CCD4a* was introduced into different chrysanthemum cultivar groups via multiple independent hybridisation events (Fig. 4e and Supplementary Fig. 31). The *CCD4a* genes of traditional Chinese chrysanthemums and pot chrysanthemums might have been introduced from *C. vestitum*, while the *CCD4a* genes of cut chrysanthemums might have been introduced from *C. zawadskii* (Fig. 4e). This implies that the breeding process of cut chrysanthemums has involved different artificial hybridisation events compared with those of traditional chrysanthemums and pot chrysanthemums. Interestingly, the *CCD4a* of traditional Japanese chrysanthemum clustered into an independent clade (Supplementary Fig. 31), which suggests that the flower colour trait of chrysanthemum might have arisen from an independent breeding process in Japan. Overall, multiple introductions of the *CCD4a* gene by artificial hybridisation have promoted carotenoid degradation in the ray floret petals of the original chrysanthemum, contributing to the flower colour diversity of modern cultivated chrysanthemums. The finding also supports the breeding and transmission history of chrysanthemum.

## Discussion

Polyploid genome assembly and analysis are still a great technical challenge due to high sequence similarities and homoeologous exchanges. To overcome these difficulties, we selected a specific haploid cultivated chrysanthemum as the material for sequencing, which greatly reduced complexity in assembly, annotation, and downstream analysis. PacBio long read sequencing allows resolving and assembling regions that are rich in repeats and of high genomic similarity. The generated *C. morifolium* genome has the contig N50 size of 1.87 Mb, scaffold N50 size of 303.69 Mb, and LAI of 27.99. It provides a valuable resource for evolutionary and comparative genomics studies. Our data suggest that the previously reported recent WGD event in *C. nankingense*[13], estimated at approximately 6 Mya, was more likely to be a WGT event (herein referred to as WGT-2) and was shared by all *Chrysanthemum* species (Fig. 2b and Supplementary Fig. 6). In addition, a very recent independent triploidization event (~3 Mya) was revealed in cultivated chrysanthemum (Fig. 2b). Therefore, *C. morifolium* underwent a total of at least three rounds of WGTs after the WGT-γ, plus a recent burst in LTRs (<0.5 Mya; Fig. 2d), resulting in a huge, complex genome architecture.

Polyploidy has long been recognised as a major force in plant evolution and speciation[64]. To date, the genomes of seven hexaploid plants have been sequenced[27,65,66], and all are recognised as being of allopolyploid origin. A previous study revealed that pot chrysanthemums are genetically more closely related to the wild progenitor *Chrysanthemum* species than to either cut chrysanthemums or traditional chrysanthemums[67]. Thus, the 'Zhongshanzigui' genome sheds light on the origin of cultivated chrysanthemum. Based on our current findings together with previous reports[46,68,69], we infer that the multiple identified introgressions, hybridisations, and polyploidization events have led to evolutionary reticulation in the *C. morifolium* polyploid complex and that *C. nankingense*, *C. lavandulifolium* and *C. seticuspe*, previously thought to be the progenitors of cultivated

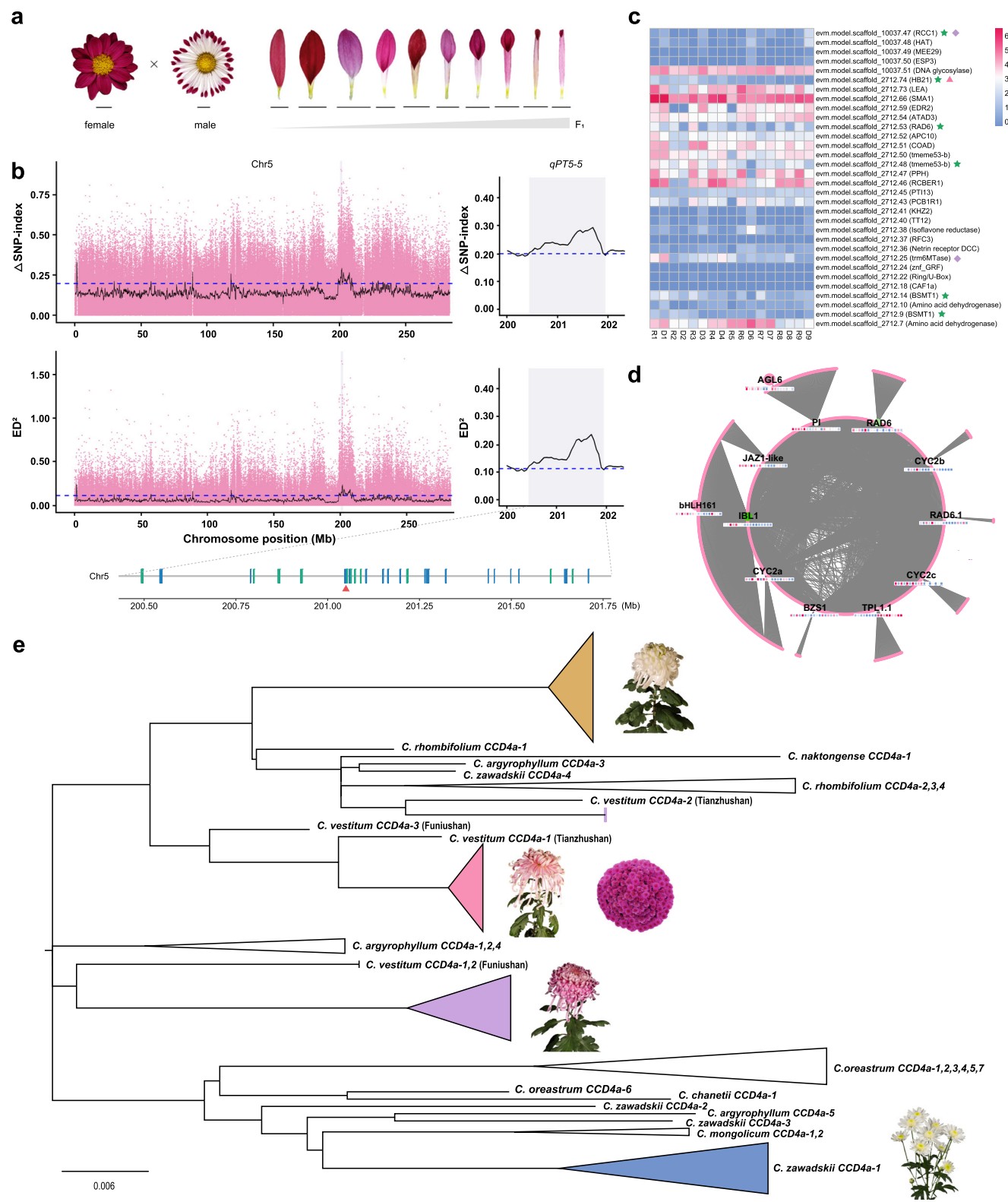

chrysanthemum, may not be the direct ancestral donors. The phylogenetic and IS analyses (Fig. 3) demonstrated that *C. rhombifolium*, which is narrowly distributed in Wushan (Chongqing, China), presented a closer relationship with cultivated chrysanthemum. Questions that remain to be further explored include the possible involvement of *C. rhombifolium* in the formation of cultivated chrysanthemum and when and how the hybridisation mixture expanded its range. The multiple lines of evidence provided here combined with

previous studies[40,51,70,71] imply that *C. morifolium* cv. 'Zhongshanzigui' is likely to be a specific segmental allopolyploid (as defined by Stebbins et al.[72].) and reveal its ongoing divergent evolution. Unfortunately, we failed to assign the 27 pseudochromosomes to subgenomes, and our conjecture of the "AA'B" genomic composition of *C. morifolium* remains to be investigated. Interestingly, we found that only ~22% of chrysanthemum homoeologs show balanced expression, which is considerably lower than the percentage in allopolyploid wheat

**Fig. 4 | The *C. morifolium* genome provides insights into diversification of flower shape and flower colour. a** Variation in corolla tube merged degree between parents and offsprings. Bar = 1 cm. **b** The QTLs controlling petal type on chromosome 5 detected by SNP-index (top) and Euclidean distance (bottom) algorithms, using SNPs. The top 1% threshold is indicated by the horizontal blue dotted line. One significant peak designated as *qPT5-5* with 32 genes is shaded and partial enlarged on the right. The green and blue lines represent the gene located on the sense strand and antisense strand, respectively. *CmRAD6* (evm.model.scaffold_2712.53) is indicated by a red triangle. **c** Expression of the 32 genes in the ray (R) and discoid (D) florets of three flat petal type cultivars (R/D1-3), three tubular petal type cultivars (R/D4-6) and three spoon petal type cultivars (R/D7-9). Green

star, pink triangle, and purple diamond, respectively, represent the gene is differentially expressed in groups of flat vs tubular, flat vs spoon, tubular vs spoon. **d** Subnetwork for the genes in turquoise module. The heatmap of the hub gene is same as that in c panel. **e** Phylogenetic tree of *CCD4a* genes from representative non-yellow wild and cultivated chrysanthemums showing the potential breeding process on flower colour. The orange triangle is merged by all the traditional Japanese chrysanthemums and one traditional Chinese chrysanthemum cultivar. The pink triangle presents a mixture of traditional Chinese chrysanthemums and pot chrysanthemums. The purple and blue triangles represent the group of traditional Chinese chrysanthemums and cut chrysanthemums, respectively. See Supplementary Fig. 31 for the full version of this tree.

(72.5%)[73] and *Brassica juncea* (83.8%)[74]. This may be explained by the rapid evolution of cultivated chrysanthemum and its specific asexual reproduction.

Cultivated chrysanthemum possesses greater morphological diversity than other *Chrysanthemum* species, especially with respect to flower shape and colour. In this study, we inferred that the time period extending back 6 Mya was an important period in the speciation of *Chrysanthemum* and the tachytelic evolution of cultivated chrysanthemums. The expansion of the MADS-box gene family and *CYC2* genes because of the recent WGT-2 and triploidization events might be responsible for the diversity of flower morphology. Based on the reference genome, we were able to preliminarily reveal the variation underlying genetic diversity of cultivated chrysanthemum (Supplementary Data 13) and identify several candidate genes involved in petal development, which will allow further fine mapping and functional verification in chrysanthemum. Moreover, our phylogenetic study of *CCD4a* provides a perspective on the independent breeding history of each cultivated type, even among different cultivars. In addition, we reconstructed the anthocyanin and flavonol biosynthesis pathways for flower colouration in chrysanthemum (Supplementary Note 8 and Supplementary Fig. 30d).

In conclusion, the genomic resources we present here can help to mine hub genes governing important traits in chrysanthemum and contribute to the study of *Chrysanthemum* evolution. Additionally, the reference genome will facilitate the application of molecular marker-assisted breeding and genome editing to cultivated chrysanthemum. The allopolyploid genome of *C. morifolium* may help to guide assembly other large complex polyploid genomes with high heterozygosity and uncertain origins.

## Methods

### Plant materials and genome sequencing
We sequenced the entire genome of a previously characterized haploid ($n = 3x = 27$) line of potted and ground-type *C. morifolium* cv. 'Zhongshanzigui' ($2n = 6x = 54$), derived from the in vitro culture of nonfertilized ovules[23] (Supplementary Fig. 1). Genomic DNA was extracted from young leaf tissue using a DNAsecure Plant Kit (TIANGEN, Beijing, China) for Illumina short read sequencing, PacBio long read sequencing, 10X Genomics and Hi-C library construction and sequencing, as described in Supplementary Note 1. Total RNA was extracted from various organs using an RNA extraction kit (Huayueyang, Beijing, China) for sequencing (RNAseq) to support genome annotation. All plant materials used in this study were collected from the Chrysanthemum Germplasm Resource Preserving Centre (Nanjing Agricultural University, Nanjing, Jiangsu, China).

### Chromosomal assembly and validation
The de novo assembly of the long reads obtained from the PacBio SMRT Sequencer was performed using FALCON[75] with the following parameters: length_cutoff_pr = 11,000, overlap_filtering_setting = --max_diff 500 --max_cov 500. The initial assembly was polished using Quiver[76] based on PacBio long reads. Then, 10X Genomics reads were aligned to

the assembly by BWA-MEM[77] using the default settings. Scaffolding was performed by FragScaff[78] with the barcoded sequencing reads. We used Pilon[79] to perform error correction based on the Illumina sequences. Subsequently, the Hi-C sequencing reads were used to align to the assembled scaffolds by BWA-MEM[4], and these scaffolds were clustered and reordered using ALLHiC[80]. Scaffolds were fine-tuned, discordant contigs were removed from scaffolds using Juicebox[81], and the final chromosome assembly was generated after careful manual inspection. The top 10× longest PacBio reads were mapped to the genome by minimap2 with the parameters: "-k 15 -w 10 -I 9 G" to assess its continuity by calculating the unique mapping (≥80% alignment length) proportion. In addition, we also present a chromosome-scale genome assembly for diploid *C. nankingense* as described in Supplementary Note 2. Synteny analysis between the genomes of diploid *C. nankingense* and *C. morifolium* was performed by MCScan (Python-jcvi).

CEGMA (http://korflab.ucdavis.edu/dataseda/cegma/)[82] and BUSCO (http://busco.ezlab.org/)[83] with the Embryophyta odb10 database and LTR Assembly Index (LAI)[84] were used to assess the completeness and continuity of the genome assembly. Gene completeness was evaluated by mapping ESTs to the assembly using BLAT (http://genome.ucsc.edu/goldenpath/help/blatSpec.html).

### Genome annotation
We adapted a combination of protein homology-based, de novo and transcriptome-based predictions to annotate protein-coding genes. The protein sequences from six species (*Arabidopsis thaliana, Daucus carota, Helianthus annuus, Lactuca sativa, Solanum lycopersicum* and *Solanum tuberosum*) were aligned to the *C. morifolium* genome using TBLASTN[25], with an *E*-value cut-off of 1e-5. GeneWise[84] was employed to predict the exact gene structure of the corresponding genomic regions in each BLAST hit. For transcriptome-based prediction, RNA-seq data from three tissues were mapped to the *C. morifolium* genome using TopHat (--splice-mismatches 2 --max-intron-length 500000 --min-intron-length 50) and Cufflinks[85] (--max-intron-length 500000 --min-intron-length 50 --max-mle-iterations 5000). In addition, Trinity[86] was used to assemble the RNA-seq data (--min_glue 2 --min_kmer_cov 2), followed by the application of PASA software[87] to improve the gene structures. The generated gene set was denoted PASA-T-set and was used to train ab initio gene prediction programs. Five ab initio gene prediction programs, namely, Augustus (version 2.5.5)[88], GENSCAN (version 1.0)[89], GlimmerHMM (version 3.0.1)[90], Geneid[91], and SNAP[92], were used to predict coding regions in the repeat-masked genome with the default parameters. Finally, all the predictions were combined using EVidenceModeller (EVM)[93] to produce the nonredundant gene set.

The functional annotation of protein-coding genes was achieved by performing BLASTp searches (*E*-value ≤ 1e-5)[94] against the SwissProt (http://web.expasy.org/docs/swiss-prot_guideline.html), NR (ftp://ftp.ncbi.nih.gov/blast/db/), InterPro (http://www.ebi.ac.uk/interpro/, V32.0), Pfam (http://pfam.xfam.org/, V27.0) and KEGG databases (http://www.kegg.jp/kegg/kegg1.html).

Transposable elements (TEs) in the *C. morifolium* genome were identified through a combination of de novo and homology-based

approaches using several databases and software packages, including RepeatModeler[95], LTR_FINDER[96], RepeatMasker and RepeatProteinMask[97], as described in Supplementary Note 2. The LTR retrotransposons were annotated with a pipeline that uses LTRharvest[98], LTR_FINDER[96] and LTRdigest[99]. We extracted the sequences of the long terminal repeats for each LTR, aligned them with MUSCLE[100], and then calculated the distance ($K$) between LTRs with the Kimura two-parameter approach. The LTR insert time was estimated using the formula

$$T = K/(2 \times r) \qquad (1)$$

where $r$ refers to a general substitution rate of $1.3 \times 10^{-8}$ per site per year in the Asteraceae family. Additional details are available in Supplementary Note 3.

## Genome evolution

To infer phylogenetic trees, we first generated a consensus gene set to represent the monoploid genome of *C. morifolium*. Protein sequences from each set of three homoeologous chromosomes were extracted for use as queries against each other, and after c-score filtering (>0.7) by JCVI (https://github.com/tanghaibao/jcvi), MCScanX[101] (-a, -e:1e-5, -u:1, -s:5) was used to determine synteny blocks with at least five gene pairs per block. The longest genes among each set of orthologous gene pairs together with the chromosome-unique genes constituted the final *C. morifolium* consensus gene set.

The protein sequences of *C. morifolium*, *A. annua*, *A. coerulea*, *A. trichopoda*, *C. canephora*, *C. cardunculus*, *C. nankingense*, *C. seticuspe*, *D. carota*, *H. annuus*, *L. sativa*, *O. sativa*, *S. lycopersicum*, *V. vinifera* and *Z. mays* were clustered into paralogues and orthologues using the program OrthoMCL[102] with an inflation parameter equal to 1.5. The single-copy gene protein sequences were independently aligned by MUSCLE[100] and then concatenated into a superalignment matrix. The phylogenetic tree of the 15 species was constructed by the ML method using RAxML[103] with 100 bootstrap replicates. The MCMCTree program implemented in Phylogenetic Analysis by Maximum Likelihood (PAML)[104] was applied to infer divergence times based on the phylogenetic tree with the following main parameters: burn-in of 10,000, sample number of 100,000, and sampling frequency of 2. In addition, the analysis of gene family expansion and contraction was performed by the CAFÉ program[105].

Synonymous substitutions per synonymous site ($Ks$) values were calculated for each gene pair in the aligned blocks. The distributions of all $Ks$ values were plotted to infer whole-genome duplication or speciation events that occurred during the evolutionary history. The peak $Ks$ values were converted to divergence times according to the formula

$$T = Ks/2\lambda \qquad (2)$$

by using a substitution rate of $8.25 \times 10^{-9}$ for asterids[13]. The dot plots between *C. cardunculus* and *C. morifolium* as well as the *C. nankingense* genome were generated with Quota synteny alignment software to visualise palaeopolyploidy events. Additional details are available in Supplementary Note 4.

## Gene family identification

The protein sequences of the ABCE floral development genes in *Arabidopsis thaliana* and reported *CYC2* genes in chrysanthemum were used as queries, and the candidates in the *C. morifolium* reference genome were identified using BLASTp with a threshold of *E*-value ≤ 1e-5. The aligned hits with at least 50% coverage of seed protein sequences and >50% protein sequence identity were selected as homologues. Then, the domains of these homologues were predicted by PFAM (http://pfam.xfam.org/). Only those genes that had the same protein domain were considered to be homologues. The results of the

identification of TPS and MADS-box family members in eight plant species, including *C. morifolium*, *A. annua*, *C. cardunculus*, *C. nankingense*, *C. seticuspe*, *L. sativa*, *H. annuus* and *A. thaliana*, are provided in Supplementary Note 3.

## The origin of cultivated chrysanthemum

For whole-genome resequencing, 12 Chinese wild *Chrysanthemum* accessions were chosen, representing all the extant potential diploid and tetraploid ancestors of cultivated chrysanthemum. Based on the filtered SNPs, the phylogenetic relationships were evaluated by constructing a maximum-likelihood tree and calculating identical score (IS) values. Gene retention analysis was performed using the chromosome-scale *C. nankingense* and *C. seticuspe*[15] genomes as a reference. To further clarify the genome structure of *C. morifolium*, two kmer-based methods were adopted, and the meiosis behaviours of the sequenced haploid and doubled haploid plants were examined. A FISH analysis was conducted to assess sequence differences among the homoeologous chromosomes. All details are available in Supplementary Note 5.

## Mining of quantitative trait loci and candidate genes for flower shape

Based on the reference genome, we were able to identify quantitative trait loci and candidate genes by integrated BSA-seq and WGCNA, as described in Supplementary Note 7. Briefly, for BSA-seq analysis, two extreme DNA pools, flat-bulk (BF) and tubular-bulk (BT), were constructed in an F1 population derived from 'Hongxiao' and 'Q5-12' by mixing equal amounts of DNA from 20 individuals for each bulk (Supplementary Fig. 25). Four libraries, including two parents and two progeny pools, were sequenced on the DNBseq-T7 platform to obtain 150 bp PE reads with an average coverage of 30×. GATK software[106] was used for SNP/InDel variant calling. Two association analysis methods, SNP-index[107] and Euclidean distance (ED)[107], were used. The top 1% of absolute ΔSNP index and ED[75] values were considered to be strongly associated with the corolla tube merged degree (CTMD).

For WGCNA, the ray florets and disc florets of nine representative cultivars were dissected in the early bloom stage for RNA extraction and transcriptome sequencing. Differential gene expression analysis was performed by DESeq[108]. After filtering low-abundance (FPKM ≤ 1) genes of samples, 74,074 ray-specific genes were used for coexpression network analysis by the WGCNA package[109] using a dynamic tree cut algorithm with a minimum module size of 50 genes and a merging threshold of 0.25. The CTMD values were used as phenotypic data to identify petal type-related modules. Coexpression networks were visualised in Cytoscape software[110].

## Phylogenetic analysis of *CCD4a* genes

To obtain the *CCD4a* nucleotide sequences of *Chrysanthemum* species, the primer set *CCD4a*-ORF-F (5'-ATGGGCTCTTTTCCCACATCT-3')/R (5'-ATAATCAAAGCGTTGTTAGGTCATT-3') was designed based on the *CCD4a* open reading frame (ORF) sequence. The PCR mixture consisted of 17 µL ddH2O, 25 µL 2× Phanta Max Buffer, 1 µL dNTP Mix, 2 µL of each *CCD4a*-ORF-F/R (Sangon Biotech, Shanghai, China), 1 µL Phanta Max Super-Fidelity DNA Polymerase (Vazyme Biotech, Nanjing, Jiangsu, China), and 2 µL DNA. The PCR program consisted of 3 min at 95℃, followed by 35 cycles of 15 s at 95℃, 15 s at 55℃, and 120 s at 72℃, followed by a 5 min extension at 72℃. The PCR product was inserted into the pEASY-Blunt Simple Cloning vector (TransGen Biotech, Beijing, China) and then transformed into DH5α for sequencing. MUSCLE software was employed for multisequence alignments[100], and a phylogenetic tree was constructed with MEGA X using the ML method[111]. Internal branching support was estimated using 1000 bootstrap replicates.

## Reporting summary

Further information on research design is available in the Nature Portfolio Reporting Summary linked to this article.

## Data availability

The raw sequencing data generated in this study have been deposited in the NCBI under accession PRJNA796762 and PRJNA895586 The chloroplast and mitochondrial genome were also available at GenBank under the accession number OP104251 and OP104742 respectively. The assembled genome sequences and annotations are available at Figshare [https://doi.org/10.6084/m9.figshare.21655364.v2]. The Arabidopsis ABCE and chrysanthemum *CYC2* genes were used as query sequences for gene family identification, which are available at Figshare [https://doi.org/10.6084/m9.figshare.21610305]. Source data are provided with this paper.

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

## Acknowledgements

This research was funded by the National Natural Science Foundation of China (32230098 to F.D.C., 31930100 to J.F.J., 32030098 to S.M.C., 32172613 to H.B.W., 32172609 to A.P.S.), the Natural Science Fund of Jiangsu Province (BK20190076 to A.P.S., BK20210395 to J.S.S.), China Agriculture Research System (CARS-23-A18 to F.D.C.), the National Key Research and Development Program of China (2021YFD1200205 to J.F.J., 2022YFF1003104 to S.M.C. 2022YFD1200504 to F.Z.), the "JBGS" Project of Seed Industry Revitalisation in Jiangsu Province (JBGS[2021]020 to F.D.C.), the European Union's Horizon 2020 research and innovation program from European Research Council (833522 to Y.V.dP.) and the Methusalem funding from Ghent University (BOF.MET.2021.0005.01 to Y.V.dP.), and a project funded by the Priority Academic Program Development of Jiangsu Higher Education Institution.

## Author contributions

F.D.C., A.P.S., J.F.J. and S.M.C. conceived and designed the study. H.B.W. prepared the genome sequencing material. A.P.S., Z.R.Z., X.T.Z. and J.S.S. led the bioinformatics analyses. F.D.C., W.M.F., S.M.C., Z.Y.G., J.F.J, F.Z. and S.Z. provided the chrysanthemum materials. H.B.W. and J.H. conducted the FISH analysis. F.Z., L.D., Y.L., D.W.J. and C.W.C. performed the BSA-seq and RNA-seq analyses. L.J.Z., J.L.Z., Z.Y.Y. and D.J.S. cloned the *CCD4* genes. J.S.S. developed the figures. A.P.S., J.S.S., Z.R.Z., X.T.Z., Y.V.dP. and F.D.C. interpreted the data and drafted the manuscript. J.S.S., A.P.S., Z.R.Z. and X.T.Z. wrote the manuscript. Y.V.dP., A.P.S., J.S.S., X.T.Z., Z.X.W., L.K.W., B.Q.D., F.C. and F.D.C. revised the manuscript. All authors read and approved the final version of the manuscript.

## Competing interests

The authors declare no competing interests.
