## [Peer Review File · Nature Communications]

Analyses of a chromosome-scale genome assembly reveal the origin and evolution of cultivated chrysanthemumReviewers' Comments:

Reviewer #1:

Remarks to the Author:

The manuscripts described about de novo whole genome assembly in *Chrysanthemum morifolium*. The species is hexaploids, however, haploid (3X) material was used in genome assembly. The manuscript covered a wide range of analysis of genomic study in *Chrysanthemum*, including gene prediction annotation, phylogenetic analysis with other *Chrysanthemum* and *Asteraceae* species, gene expression analysis and genetic basis analysis for investigation of flower evolution. The manuscript is generally well written, but sometimes observed lack of explanation, that makes the reliability of the results.

In the analysis of ploidy genomes, homoeologous sequences often interfere with accurate analysis in assembly, gene prediction, gene expression, and variant detection. However, this paper did not describe any special considering in analysis against obstructions caused by polyploidy. Meanwhile, the results were very clear. Therefore, it was concerned that the undescribed data analysis was processed to present better results than the original. For example, in the Hi-C analysis of ploidy species, sequences are usually mapped to chimeras on homoeologous sequences, which makes it difficult to obtain accurate chromosome-scale sequences. Therefore I request that authors add descriptions about the way to overcome the obstacles in polyploid analysis.

The authors suggested that the structure of the cultivated gymnosperm may be AAB, however, the assembly results rather showed that the sequenced material is complete allo-triploid. Because each homoeologous chromosome were clearly distinct. If the sequenced material used in this study was considered as allo-triploid, the methods used in variant call and RNA-Seq was accepted. However, if the sequenced material were AAB or a fully auto-polyploidy, as many of chrysanthemum studies suggested, the reliability of results in variant call and RNA-Seq were questionable because accurate analysis may be hampered by mapping homoeologous sequences across chromosomes. Therefore, the authors should determined whether the material was allo- or auto- polyploidy, and change the analysis methods depending on their decision.

Nature Communication is a high impact journal, and it is requested obvious novelty in science, The 'first report' in genome assembly is not enough, description about the scientific novelty in the genotoxic analysis of auto-polyploidy species should be clearly stated in the manuscript.

Minor comments entire the manuscript

Properly use the words, homologous and homoeologous. I think descriptions using 'homologous' should be replaced with 'homoeologous', in the case that the authors described homologous sequences among the homoeologous chromosomes.

L77-78 The description is incorrect. The genome assembly reported by Nakano et al, is chromosome-scale, not fragmented.

L100

The genome size of the haploid ($n=3x=27$) cultivated chrysanthemum cv. 'Zhongshanzigui' ($2n=6x=54$; Supplementary Fig. 1)

I am confused about this sentence. Is the description for haploid or diploid? Based on the description of estimated genome size, I guess that the result is for haploid. If so, what does 'heterozygosity' mean? Does it mean 'sequences differences between homoeologous sequences'?

L104–106; Specify type of reads using polishing in the sentence.

L106-L108: 10X Genomics data: should be described as 'genome sequences derived from a 10X Genomics library'

Hi-C data: should be described as 'genome sequences derived from a Hi-C library'

Because a triploid material was sequenced in this study, homoeologous sequences would be collapsed or classified to haplotig sequences by Falcon unzip. Please show the assembly result of Falcon unzip, i.e. total length and assembled sequences of primary and haplotig sequences in a Supplementary table.

Did you use both primary and haplotype assembly for further scaffolding with 10X Genomics reads? Please specify.

Supplementary Table 2.

The insert size of Hi-C should not be 350 bp.

Supplementary Table 4. According to Supplementary Table 3, the number of contigs was 19,524, while the total contig number in Table 4 was 19,509. Why were the numbers different?

L108-109: Were the scaffolds chimeric and located across chromosomes or correctly phased? Please show the number of scaffolds mapped each chromosome in Table 4.

L115 hexaploid nature of cultivated chrysanthemum

The description is unclear. What does 'hexaploid nature' mean here? Did the authors intend to say that the 'Zhongshanzigui' is all-polyploid because the three homoeologous chromosomes were clearly distinguished?

L123-125 and Supplementary Table 6.

Was BUSCO analysis performed for scaffolds or chromosome-scale sequences? Please specify. I expected that most of the BUSCOs were identified as double. However, 10% BUSCOs were identified as single. Does it mean that some chromosomes don't have a part of BUSCOs genes?

Supplementary Table S7.

Show the source sequences (reference or NCBI ID) of *C. seticuspe*, *Artemisia annua*, *Helianthus annuus*, *Cynara cardunculus*, *C. makinoi*, *C. lavandulifolium* and *C. seticuspe*.

L125 indicating higher integrity than that of the released genomes of other Asteraceae species. The results of BUSCOs of *C. seticuspe* reported by Nakano et al. (2021) was 97.4%, similar to that of the author's results.

L131-134 Pacbio reads mapping

I expected higher sequence similarity across three homoeologous chromosomes (i.e., multiple mapping). Please specify the mapping method including parameters used in this study. Please also show the evidential results of the description.

Supplementary Table S16. Note the first author and publication year of the references in the table. Supplementary Fig. 3. What does 'other ortholog' mean? Please explain.

L258 With considering the heterozygosity and polyploidy of the sequenced materials, the average depth of 8.5x are low, and doubt the reliability of the variant call. I recommend filter variants showing more than 20 DP.

L251~ Origin of chrysanthemum

Can we trust the tree? It is unreasonable that the genetic diversity in *C. indicum* was larger than interspecies. The reliability of the mutation detection used in this analysis is questionable.

I was considering that *C. morifolium* was an auto-polyploid. However, the distinguished sequences among homoeologous chromosomes shown in this study suggested a possibility that *C. morifolium* is an allo-polyploid species. If it is allo-polyploid, the wild progenitor would not be a single species. Therefore, in the case the authors consider that *C. morifolium* was allo-polyploid species, I

recommend making a phylogenetic tree in each chromosome, to confirm the number of possible progenitor species.

L271-L274. It seems Supplementary Fig 14 does not show evidence of small introgression. How can we find the small integration from the figure?

L275-L284

I think the results suggested in Supplementary Fig 15 did not deny the possibility that *C. nankingense* is a progenitor of *C. morifolium*, and rather suggest that

- the sequenced material was not fully auto-polyploid species
- *C. nankingense* would be a donor.

I recommend adding *C. seticuspe* genome reported by Nakano et al. (2021), as reference of the comparison between *C. nankingense* and *C. morifolium*.

L305 I think Supplementary Fig. 17 rather suggested a possibility of AAB, not fully auto-polyploidy species.

Chapter Homoeolog expression bias in hexaploid chrysanthemum
Supplementary Fig 19

I could not find the description about the organs extracted RNAs for the analysis in Supplementary Note 6. Please add details of the process in transcriptome sequencing. In addition, I could not understand how authors identify the chromosomes that express each gene, when homoeologous genes were located across chromosomes.

Supplementary Fig 20. What dose G1d~G3s mean?

Reviewer #2:

Remarks to the Author:

In the presented article, titled, "A chromosome-scale genome assembly of hexaploid cultivated chrysanthemum", Song et al., assembled a highly heterozygous and economically important ornamental plant, *Chrysanthemum morifolium* Ramat.. Authors managed to assemble a difficult genome, and Hi-C contact map supported their claim of a nice genome resource. Authors identified a recent WGT event, and also suggested allopolyploid nature of *Chrysanthemum* genome with AAB composition. Authors have used strong evidence while adopting enough caution to claim WGT, allopolyploid nature of the genome, which is highly admirable. I find the established resource from this study important step to explore and improve this valuable plant species, and was convinced with all the claims that authors made based on the described evidence. There are few minor issues that I hope authors could resolve (mentioned below-

1. Line 108. In my understanding, Hi-C data orders the scaffold, but the sequencing data itself is not integrated. The sentence here is providing a different meaning. Please consider rephrasing it.

2. Although Hi-C contact map does show 9 groups, each group with 3 elements, thus n=27, I wonder if this species has been used for karyotyping? If they could show the chromosome numbers, or provide a reference to support, that would be great. This will further validate their Hi-C data-based results.

3. Line 114, "clear cut" sounds typo or strange. Please consider rephrasing it.

4. Please provide number of gaps within this genome in the manuscript text. It will provide a sense of completeness and quality of genome. Given the heterozygosity and the repeat content of this plant genome, I feel that the contig N50 is quite decent.

5. Authors should consider comparing individual groups (9 groups that they identified) based on synteny, and should try and see if the gene order is changed, and if that has any implication for flower color or shape or other biological processes. Authors did mention differences in the expression of genes within groups (for example between chr13 and chr14), but what percentage of synteny they

observed within each groups? Is this same or similar across all groups? Does order of genes remains the same? What changes whatsoever they could observed?

6. Authors used Wild type species to ascertain ploidy nature of Chrysanthemum, and reported 104597 synonymous SNPs. I am curious if these SNPs had any associated with features that are involved with the coloring or shape of flowers. Compared to these wild types, is their any specific genomic regions that showed over representation of these SNPs? Any biological interpretations that one could derive from it?

7. Line 238, sentence seems very strange, some mistake. Please check and correct it.

8. Please provide your interpretation on different copia-to-gypsy ration that they observed in discussion section.

9. The color of flowers certainly involved several secondary metabolic pathways including phenylpropanoid among others. Authors should consider describing this aspect using the generated dataset. This will improve scope of this manuscript to a broader audience.

Overall, the genome quality and the interpretation for this manuscript is sound and impressive. The genomic resource itself is valuable, and therefore, with these changes, I find this manuscript an important advancement for future studies on Chrysanthemum including species improvement and novel biology based discoveries.

Reviewer #3:

Remarks to the Author:

Chrysanthemum is one of the most widely cultivated ornamental plants, with great economic and cultural values. For a long time, the various ploidy has limited the research for chrysanthemum species, especially under their complex evolutionary history (e.g., the frequent interspecific hybridizations). Due to the limitation of sequencing and assembling technologies, it is really difficult to generate the high-quality genome assembly for the polyploid species. Even now, the reported genomes of hexaploidy species are quite rare. In this manuscript, Song et al. reported a newly assembled genome of hexaploid cultivated chrysanthemum, *Chrysanthemum morifolium* Ramat., explored its evolutionary history (including the genome structure, WGD and WGT events, and origin history), and further identified the novel genes controlling key ornamental traits with multiple evidences. In summary, this is an exciting work, which provides an extremely valuable genetic resource for researchers in the field and could facilitate the related research (e.g., molecular breeding) for chrysanthemum species.

Overall, I am glad to review this nice work. I am pleased to recommend it for publication after the revision according to my following major and minor comments.

Major:

1. To study the origin of *C. morifolium*, the authors mapped the reads of 12 wild species to *C. morifolium* genome and further reconstructed the phylogenetic topology using their SNPs. However, the ploidy of sampled species is various, including diploid, tetraploid, and hexaploid. For phylogenetic analysis, it is unreasonable to map the samples with such various ploidy to a single reference genome and directly generate the phylogenetic tree based on such obtained SNPs. So, the deduced origin of *C. morifolium* is not credible. Actually, the common way to study the origin of polyploid species (tetraploid and/or hexaploidy species) is to divide their genomes into different subgenomes and further construct their phylogenetic relationships. Furthermore, the comprehensiveness of sampling will directly affect the result. The authors should declare whether all wild relatives are sampled in this study. It is better to provide a map of their distribution areas in the manuscript.

2. The authors argued that *C. morifolium* is an allopolyploid with the genome structure of "AAB" via multiple evidences. However, when mapping to the reference genome, a pattern of uniform coverage of different wild species was observed. It is strange and unreasonable. If *C. morifolium* is an

allopolyploid ("AAB"), when mapping diploid or tetraploid species to it, we could expect an obvious different coverage depth between A and B subgenomes. Otherwise, if the methods for read mapping applied correctly, it suggests *C. morifolium* is more likely to be an autopolyploidy ("AAA"). The authors should justify these conflicts.

3. The authors amplified different alleles of CCD4a and speculated they might contribute to the flower color diversity of modern cultivated chrysanthemums. It is understandable that, for such hexaploidy non-model plant species, it is hard to perform the *in vivo* functional verification tests. However, more evidence (e.g., data from the transcriptomes) should be provided to confirm such inference. Otherwise, it needs turn down the conclusion.

Minor:

1. L35: Change "Million years ago" to "million years ago".
2. L128: Please provide the full Latin name of *E. chinocloa* when it occurs for the first time.
3. L177: Change "Functional annotation" to "Functional enrichment analysis".
4. L192-L210: The naming system of WGD/WGT ("WGT-1", "WGT-γ", "WGD-2", and "WGT-2" et al.) is really hard to follow. I suggest the authors to label them in Fig. 2a on corresponding positions at the same time. Moreover, why there is no "WGD-1"?
5. L258: Whether the 12 samples used in this study are different species. I notice some of them look like different populations/ecotypes/varieties. The statistics of 12 samples should be provided in Supplementary Tables, e.g., their sampled locations, sequencing data size, mapping coverage, and mapping depth et al.
6. L263: According to the Supplementary Note, the SNPs used for phylogenetic analysis are 4DTv sites. If so, please change "synonymous" to "4DTv". More details should be provided in the Methods or Supplementary Notes. For example, the authors should declare why the analysis was based on 4DTv sites, but not the genome-wide SNPs or intergenic SNPs.
7. L264-L265: As I mentioned in the Major Comment 1, I do not think the phylogenetic relationships constructed via such way is believable. In addition, based on this phylogenetic tree (Fig. 3a), *C. dichrum* and *C. indicum* (Hubei) may be the same closely to *C. morifollum* as *C. indicum* (Nanjing).
8. L268-L272: Why the rank of identity score (IS) is not consistent to the phylogenetic tree (Fig. 3a). For example, *C. rhombifolium* is the most closely to *C. morifollum* in Fig. 3a, but the IS of *C. rhombifolium* (0.832) is less than that of *C. indicum* from Nanjing (0.859), Tianzhushan (0.840) and Hubei (0.837), and *C. dichrum* (0.834). I am sorry to say it again, does it also reflect the method applied for phylogenetic analysis is unreasonable, as I mentioned in Major Comment 1. Please justify it.
9. Methods & Supplementary Notes: More details should be provided in the Methods and/or Supplementary Notes. There is so less information on how the authors performed the analyses. The readers could not know the details of methods and thus could not further evaluate the accuracy of some results. Please improve the sections of Methods & Supplementary Notes.
10. L561-L569: The list of query genes and the information of the domains should be provided.
11. L599: I believe the editor would confirm all data has been released before publication.
12. Fig. 2a. All fossil calibrations used in the study should be listed and cited.
13. Fig. 3a: Please confirm whether the phylogenetic tree is constructed by ML methods. TreeBest is always applied in phylogenetic analyses based on NJ methods. The ploidy labeled on the right is too small.
14. Fig. 3c & Supplementary Fig. 8: It is difficult to observe the differences between the average values of different groups. The statistical test could be performed, or the figure could be improved.
15. Fig. 4c: Please annotate and change the gene ID on the right to the gene names.
16. Supplementary Fig. 5: The unit of the P value in the figure (the heat map) should be labeled.
17. Supplementary Fig. 9: Typo? Should the unit of P-adjust value be Log10 or -Log10?
18. Table 1: Typo. Change "BUSCO completeness of annoation" to "BUSCO completeness of annotation".

Detailed Response to Reviewers

Reviewer #1 (Remarks to the Author):

The manuscripts described about de novo whole genome assembly in *Chrysanthemum morifolium*. The species is hexaploids, however, haploid (3X) material was used in genome assembly. The manuscript covered a wide range of analysis of genomic study in *Chrysanthemum*, including gene prediction annotation, phylogenetic analysis with other *Chrysanthemum* and Asteraceae species, gene expression analysis and genetic basis analysis for investigation of flower evolution. The manuscript is generally well written, but sometimes observed lack of explanation, that makes the reliability of the results.

Response: Thanks for the positive comments on our manuscript. Given the tremendous complexity of the cultivated chrysanthemum genome, here, we used a haploid plant as sequence material that was previously obtained from the *in vitro* culture of non-fertilized ovules (Wang et al., 2014). Actually, the use of haploid material for assembly is common in genome sequence projects to reduce the complexity of assembly and has been used in many other plant genome sequencing projects, such as rose (Hibrand Saint-Oyant et al., 2018) and sugarcane (Zhang et al., 2018). Nevertheless, the estimated large genome size (8.47~9.02 G) as well as the high levels of heterozygosity (0.71%, please see our response to the third 'minor' comment) and repeats (88.85%) indicate its complex nature, which consumed us a great deal of time and computational resource to complete the high-quality *C. morifolium* genome assembly.

To avoid confusion or misunderstanding, we have deleted the 'hexaploid' in both title and throughout the revised manuscript, where required, and also added several explanations. Overall, we have made substantial revisions to improve the quality of our manuscript according to the reviewers' comments. The findings provided here would give further insights into the evolutionary history of chrysanthemum and will facilitate the in-depth research in identification of novel genes underlying important horticultural traits through forward genetics approaches and accelerate molecular breeding of chrysanthemum. The *C. morifolium* genome might also provide a valuable reference for other complex polyploids that are not of strict allo- or auto-polyploid origin.

References:

Wang, H. et al. Characterization of in vitro haploid and doubled haploid *Chrysanthemum morifolium* plants via unfertilized ovule culture for phenotypical traits and DNA methylation pattern. *Front. Plant Sci.* 5, 738 (2014).

Hibrand Saint-Oyant, L. et al. A high-quality genome sequence of *Rosa chinensis* to elucidate ornamental traits. *Nat. Plants* 4, 473-484 (2018).

Zhang, J. et al. Allele-defined genome of the autopolyploid sugarcane *Saccharum spontaneum* L. *Nat. Genet.* 50, 1565-1573 (2018).

In the analysis of polyploid genomes, homoeologous sequences often interfere with accurate analysis in assembly, gene prediction, gene expression, and variant detection. However, this paper did not describe any special considering in analysis against obstructions caused by polyploidy. Meanwhile, the results were very clear. Therefore, it was concerned that the undescribed data analysis was processed to present better results than the original. For example, in the Hi-C analysis of ploidy species, sequences are usually mapped to chimeras on homoeologous sequences,

which makes it difficult to obtain accurate chromosome-scale sequences. Therefore I request that authors add descriptions about the way to overcome the obstacles in polyploid analysis.

Response: To overcome the added complexities caused by polyploidy, we selected a specific haploid strain of the hexaploid cultivated chrysanthemum for sequencing, as described in the previous response. For the assembly, we used high sequencing depth data, i.e., 1,022.3 Gb (120.70×) PacBio continuous long reads and 1,002.9 Gb (~118.50×) Hi-C data, comparable to what has been used for other reported diploid Asteraceae genomes and other polyploid genomes.

Due to the huge genome size of *C. morifolium*, we spent more than 2.5 M CPU hours in genome assemble and 150 K CPU hours for mapping the Hi-C data. In the Hi-C process, we fixed various program errors. To fix switch errors and chimeras, we went through considerable effort to manually identify mis-assembled contigs that displayed abnormal long-rang contact patterns from paired-end reads alignments against the contig assembly using juicer tools. Then, continuity of the genome assembly was further assessed by mapping the top longest PacBio reads to the genome by minimap2 using the following parameters: “-k 15 -w 10 -I 9G”. In the end, we found that 82.24% of the selected reads could be uniquely mapped to only one chromosome with $\geq 80\%$ alignment length, which again is comparable to the previously reported 89% for autotetraploid cultivated alfalfa (Chen et al., 2020). Finally, we also assembled a chromosome-scale genome of diploid wild chrysanthemum to verify the accuracy of the cultivated chrysanthemum genome assembly. We have detailed the results section and added the relevant description to the revised version of our manuscript (lines 452-456 and 527-533).

Reference:

Chen, H. et al. Allele-aware chromosome-level genome assembly and efficient transgene-free genome editing for the autotetraploid cultivated alfalfa. *Nat. Commun.* 11, 1-11 (2020).

The authors suggested that the structure of the cultivated gymnosperm may be AAB, however, the assembly results rather showed that the sequenced material is complete allo-triploid. Because each homoeologous chromosome were clearly distinct. If the sequenced material used in this study was considered as allo-triploid, the methods used in variant call and RNA-Seq was accepted. However, if the sequenced material were AAB or a fully auto-polyploidy, as many of chrysanthemum studies suggested, the reliability of results in variant call and RNA-Seq were questionable because accurate analysis may be hampered by mapping homoeologous sequences across chromosomes. Therefore, the authors should determined whether the material was allo- or auto- polyploidy, and change the analysis methods depending on their decision.

Response: We understand the reviewer's concern. In fact, distinguishing between different forms of polyploidy is not always straightforward (TATE et al., 2005). Generally, chromosome behavior, fertility, segregation ratios, morphology, coupled with genetic data, are principal criteria for distinguishing between autopolyploids and allopolyploids. Based on the high-quality *C. morifolium* genome, we tried very hard to unravel the origin of chrysanthemum. According to genome architecture, molecular markers and cytological experiments provided in this study, together with evidence from previous reports in *C. morifolium*, as well as the comparisons with other typical allopolyploids and autopolyploids, we conclude that *C. morifolium* cv ‘Zhongshanzigui’ is likely a specific ‘segmental allopolyploid’, i.e., neither a strict allopolyploid nor a strict autopolyploid. Meanwhile, the estimated high level of heterozygosity (0.71%,

Supplementary Table 1), clearly distinguishable Hi-C interactions heatmap (Fig. 1b), gene retention patterns (Supplementary Figs. 15,16) and 13-mers clustering (Supplementary Fig. 18), Smudgeplot (Fig. 3c) analyses as well as the added SNP-based phylogenetic tree (Supplementary Fig. 17; please see our sixth response from the end) and FISH karyotypes (Supplementary Fig. 20; please see our response to Reviewer 2) indicate the substantial differences between the homoeologous chromosomes, reflecting the ongoing divergent evolution of *C. morifolium* and giving rise to its speculative “AA'B” genomic composition. Additionally, the phylogenetic trees (Fig. a) constructed by single-copy gene families also showed that the differentiation among homoeologous chromosomes is comparable to that in inter-species, especially for the chromosome sets Chr1-Chr2-Chr3 to Chr19-Chr20-Chr21. With regard to the RNA-seq, to accurately quantify homoeologous gene expression, only the reads that uniquely mapped were kept for analysis (Supplementary Note 6). In the revised version of our manuscript, we have now revised and further clarified the “Origin of cultivated chrysanthemum” section (lines 253-335) and the Discussion (lines 479-483).

Fig. a Phylogenetic trees showing the evolutionary relationship among each of the three homoeologous chromosomes, *C. nankingense* (Cna) and *C. seticuspe* (Cse). The nine phylogenetic trees were constructed by RAxML using 312, 350, 291, 202, 351, 350, 203, 260 and 260 single-copy gene families, respectively. *A. annua* (Aan) was used as an outgroup. Blue and red numbers at branches indicate the branch length and bootstraps, respectively.

Reference:

TATE, J. A., SOLTIS, D. E., & SOLTIS, P. S. Polyploidy in plants. In *The evolution of the genome* (pp. 371-426). Academic Press (2005).

Nature Communication is a high impact journal, and it is requested obvious novelty in science, The 'first report' in genome assembly is not enough, description about the scientific novelty in the genptomic analysis of auto-polploidy species should be clearly stated in the manuscript.

Response: Chrysanthemum is a globally important ornamental plant, renowned for its great economic, cultural, and symbolic values. Although reference genomes of many important ornamental plants have been generated, such as rose (Raymond et al., *Nature Genetics*, 2018; Hibrand Saint-Oyant et al., *Nature Plants*, 2018), orchid (Zhang et al., *Nature Genetics*, 2015; Li et al., *Nature Plants*, 2022), and waterlily (Zhang et al., *Nature*, 2019), comparable work for cultivated chrysanthemum has been lagging behind, primarily owing to its large, polyploid and highly heterozygotic genome. Here, we generated a near-complete assembly for cultivated chrysanthemum comprising 27 pseudochromosomes (8.15 Gb), with contig N50 (1.87 Mb), scaffold N50 (303.69 Mb), and LAI (27.99), and which can be considered as a golden genome standard for hexaploidy chrysanthemum. We show that the present high-quality reference genome sequence can be used to unravel chrysanthemum's polyploidization history and for identifying novel genes controlling key ornamental traits through forward genetics approaches. We provide multi-level evidence (see the previous response) that *C. morifolium* is likely to be a segmental allopolyploid, i.e., neither a strict allopolyploid, nor a strict autopolyploid. In the revised version of our manuscript, we have detailed the genome assembly and highlighted the scientific novelty (see e.g., lines 89-97, 452-456, and 501-506, of the revised version).

References:

- Raymond, O., et al. The *Rosa* genome provides new insights into the domestication of modern roses. *Nat. Genet.* 50, 772-777 (2018).
- Hibrand Saint-Oyant, L., et al. A high-quality genome sequence of *Rosa chinensis* to elucidate ornamental traits. *Nat. Plants* 4, 473-484 (2018).
- Cai, J., et al. The genome sequence of the orchid *Phalaenopsis equestris*. *Nat. Genet.* 47, 65-72 (2015).
- Li, M. H., et al. Genomes of leafy and leafless *Platanthera* orchids illuminate the evolution of mycoheterotrophy. *Nat. Plants* 8, 373-388 (2022).
- Zhang, G. et al. The *Apostasia* genome and the evolution of orchids. *Nature*, 549, 379-383 (2017).
- Zhang, L. et al. The water lily genome and the early evolution of flowering plants. *Nature* 577, 79-84 (2020).

Minor comments entire the manuscript

Properly use the words, homologous and homoeologous. I think descriptions using 'homologous' should be replaced with 'homoeologous', in the case that the authors described homologous sequences among the homoeologous chromosomes.

Response: We have replaced "homologous" by "homoeologous" throughout our revised manuscript, where required.

L77-78 The description is incorrect. The genome assembly reported by Nakano et al, is chromosome-scale, not fragmented.

Response: Corrected. Please see lines 74-77 of the revised manuscript (with track changes).

L100

The genome size of the haploid ($n=3x=27$) cultivated chrysanthemum cv. 'Zhongshanzigui' ($2n=6x=54$; Supplementary Fig. 1)

I am confused about this sentence. Is the description for haploid or diploid? Based on the description of estimated genome size, I guess that the result is for haploid. If so, what does 'heterozygosity' mean? Does it mean 'sequences differences between homoeologous sequences'?

Response: The description about the genome survey based on *K*-mer analysis was for the haploid plant *C.morifolium* cv. 'Zhongshanzigui'. We have changed this sentence to avoid misunderstanding. The 'heterozygosity' estimated by *K*-mer statistics here indeed referred to sequence differences between homoeologous sequences. However, since this might be confusing, we have deleted the description of heterozygosity. Please kindly check lines 100-101 and 103-104, of the revised manuscript.

L104—106; Specify type of reads using polishing in the sentence.

Response: We polished the initial assembly using Quiver based on PacBio long reads. We have added the information in lines 104-106 of the revised manuscript.

L106-L108: 10X Genomics data: should be described as 'genome sequences derived from a 10X Genomics library'

Hi-C data: should be described as 'genome sequences derived from a Hi-C library'

Response: We have changed these sentences.

Because a triploid material was sequenced in this study, homoeologous sequences would be collapsed or classified to haplotig sequences by Falcon unzip. Please show the assembly result of Falcon unzip, i.e. total length and assembled sequences of primary and haplotig sequences in a Supplementary table.

Did you use both primary and haplotype assembly for further scaffolding with 10X Genomics reads? Please specify.

Response: Thank you for pointing this out. We assembled the PacBio continuous long reads (CLR) into contig sequences using only the Falcon software, and the p-contig of the Falcon result was used to polished and further scaffolding without Falcon unzip. To evaluate the effect of collapse, we used the PacBio long reads mapping to the *C. morifolium* genome assembly. The average depth is $9\times$, we stat the 2-fold depth ($18\times$) region. The result showed that there was $\sim 1.8\%$ (148.4 M/8154.3M) of the assembly may be collapsed. So, we considered the proportion of collapse in our assembly is very small, although we used falcon. We have revised the relevant description in lines 104-106 and 521-522 of the revised manuscript.

The insert size of Hi-C should not be 350 bp.

Response: Corrected.

Supplementary Table 4. According to Supplementary Table 3, the number of contigs was 19,524, while the total contig number in Table 4 was 19,509. Why were the numbers different?

Response: Sorry for the mistake. The contig number of Chr0 in Supplementary Table 4 should be '6,621' rather than '6,246', thus the total number of contigs was 19,524. Corrected.

L108-109: Were the scaffolds chimeric and located across chromosomes or correctly phased ?

Please show the number of scaffolds mapped each chromosome in Table 4.

Response: Yes. We have added the number of scaffolds mapped on each chromosome to Supplementary Table 4.

L115 hexaploid nature of cultivated chrysanthemum

The description is unclear. What does 'hexaploid nature' mean here? Did the authors intend to say that the 'Zhongshanzigui' is all-polyploid because the three homoeologous chromosomes were clearly distinguished?

Response: The Hi-C interactions heatmap (Fig. 1b) clearly distinguishes the three pseudochromosomes within a homoeologous group, indicating the sequence differences among the homoeologous chromosomes. However, we do not think this alone proves that 'Zhongshanzigui' is certainly a fully allopolyploid. We have revised this sentence in lines 113-114.

L123-125 and Supplementary Table 6.

Was BUSCO analysis performed for scaffolds or chromosome-scale sequences? Please specify.

Response: The BUSCO analysis was performed based on the chromosome-scale sequences of *C. morifolium*. We have revised both the text of the manuscript (line 123) and the Supplementary Table 6.

I expected that most of the BUSCOs were identified as double. However, 10% BUSCOs were identified as single. Does it mean that some chromosomes don't have a part of BUSCOs genes

Response: We understand the reviewer's concern. It is a common situation that massive gene loss occurred after the WGD or WGT event in plant evolutionary history (Sankoff & Zheng, 2018). Here, we found *C. morifolium* underwent a very recent polyploidization event (~ 3Mya), which might lead to the slightly higher proportion of single-copy BUSCOs. The single-copy and duplicated BUSCOs for autotetraploid alfalfa (Chen et al., 2020) and autopolyploid sugarcane (Zhang et al., 2018) were 7.05% and 90.11%, 14.40 % and 81.00%, respectively, which is comparable to our results. Nevertheless, the high BUSCO completeness of 97.70%, as well as other parameters presented in the manuscript, suggest that the assembly and annotation of the *C. morifolium* genome is of high-quality.

References:

Sankoff, D., & Zheng, C. Whole genome duplication in plants: implications for evolutionary analysis. In *Comparative Genomics* (pp. 291-315). Humana Press, New York, NY (2018).
Chen, H. et al. Allele-aware chromosome-level genome assembly and efficient transgene-free genome editing for the autotetraploid cultivated alfalfa. *Nat. Commun.* 11, 1-11 (2020).
Zhang, J. et al. Allele-defined genome of the autopolyploid sugarcane *Saccharum spontaneum* L. *Nat. Genet.* 50, 1565-1573 (2018).

Supplementary Table S7.

Show the source sequences (reference or NCBI ID) of *C. seticuspe*, *Artemisia annua*, *Helianthus annuus*, *Cynara cardunculus*, *C. makinoi*, *C. lavandulifolium* and *C. seticuspe*.

Response: Added.

L125 indicating higher integrity than that of the released genomes of other Asteraceae species. The results of BUSCOs of *C. seticuspe* reported by Nakano et al. (2021) was 97.4%, similar to that of the author's results.

Response: Yes. We have revised this sentence (lines 124 of the revised manuscript).

L131-134 Pacbio reads mapping

I expected higher sequence similarity across three homoeologous chromosomes (i.e., multiple mapping). Please specify the mapping method including parameters used in this study. Please also show the evidential results of the description.

Response: Thanks for pointing this out. Based on the method mentioned in the article of cultivated alfalfa (Chen et al., 2020), the top 10x longest PacBio long reads were mapped to the *C. morifolium* reference genome by minimap2 with following parameters: “-k 15 -w 10 -I 9G”, and we found that 82.24% of the selected reads could be uniquely mapped to only one chromosome with $\geq 80\%$ alignment length. We have added this information in the revised Methods section (see lines 528-530).

Reference:

Chen, H. et al. Allele-aware chromosome-level genome assembly and efficient transgene-free genome editing for the autotetraploid cultivated alfalfa. *Nat. Commun.* 11, 1-11 (2020).

Supplementary Table S16. Note the first author and publication year of the references in the table.

Response: Thanks. We have added the information of the first author and publication year to the table.

Supplementary Fig. 3. What does ‘other ortholog’ mean? Please explain.

Response: The ortholog was categorized according to routine comparative genomics analysis. Here, we taken the species A for example to explain the ortholog classification (Table a). Each species of this gene family must have at least one gene, if there is only one member within this gene family for species A, it is regarded as a single-copy ortholog, else if there is more than one member within this gene family for species A, it is regarded as a multiple-copy ortholog. Unique ortholog indicates species-specific gene family. Besides above, the number of species having at least one gene family member smaller than the number of species used for comparative genomics

analysis species number, if there is at least one member within this gene family for species A, it is regarded as other ortholog. We have added the explanation in the legend of Supplementary Fig.3.

Table a The gene family classification for species A

Family ID	Species A	Species B	Species C	Classification for species A
1	1	2	7	Single-copy ortholog
2	3	1	2	Multiple-copy ortholog
3	1	0	0	Unique ortholog
4	5	0	0	Unique ortholog
5	1	0	2	Other ortholog
6	2	1	0	Other ortholog

L258 With considering the heterozygosity and polyploidy of the sequenced materials, the average depth of 8.5x are low, and doubt the reliability of the variant call. I recommend filter variants showing more than 20 DP.

Response: Thank you for pointing this out. We agree with the reviewer and have updated the ML phylogenetic tree using a SNP set screening with a more stricter SNP filter criteria (depth > 20 & MAF > 0.05 & miss < 0.1). The reconstructed species phylogeny (Fig. 3a) was generally similar with our previous result and consistent with the IS analysis. *C. rhombifolium* and *C. indicum* (Nanjing) were still tightly clustered with *C. morifolium*. We note that *C. indicum* (Wuyishan) was separate from the other species. A similar result was found in our previous study that investigated the genetic diversity of 12 geographical populations of *C. indicum* based on ISSR and SRAP markers (Zhang et al., 2007). These findings indicate the reliability of the phylogenetic relationships. We have changed Fig. 3a and revised the descriptions in lines 261-265 and Supplementary Note 5.1.

Fig. b Dendrogram for the 12 geographical populations of *C. indicum* based on ISSR and SRAP markers by UPGMA method (Zhang et al., 2011). No. 12 denotes *C. indicum* (Wuyishan).

Reference:

Zhang, X., Zhang, F., Chen, F., Guo, H., Chen, S. Analysis of genetic diversity among 12 geographical populations of *Dendranthema indicum*. Journal of Nanjing Agricultural University 34, 48-54 (2011).

L251~ Origin of chrysanthemum

Can we trust the tree? It is unreasonable that the genetic diversity in *C. indicum* was larger than

interspecies. The reliability of the mutation detection used in this analysis is questionable.

Response: We have updated the tree as suggested. Actually, *Chrysanthemum* is a genus where the different species are generally closely related and natural interspecific hybridization is prevalent. Among them, *C. indicum* species have multiple ecotypes, different ploidies (diploid and tetraploid) and high genetic diversity. Numerous studies tried to characterize the genetic diversity among various *C. indicum* cytotypes and to elucidate the complex relationships within *Chrysanthemum* species from morphological (Wang et al., 1993), cytological (Taniguchi et al., 1987) and molecular data (Yang et al., 2006). Like our results, other researchers found that there was high genetic diversity within the different geographical populations of *C. indicum*, which was even larger than that within interspecies. For instance, our previous study (Ding et al., 2007) based on isozyme analysis of 27 *Chrysanthemum* (previously known as *Dendranthema*) accessions revealed that not all the *C. indicum* cytotypes clustered together (Fig. c). Dai et al. (1998) also discovered the relatively dispersed relationship among *Chrysanthemum* species based on RAPD markers, where *C. indicum* (Huangshan) (Hin) clustered with *C. nankingense* (Nan) and *C. morifolium* 'Hangju' (Han), while *C. indicum* (Tianzhusan) (Tin), together with *C. zawadskii* (Zaw), formed an independent cluster (Fig. d). We speculate that interspecific hybridization within the *Chrysanthemum* genus and *C. indicum* habitat diversity contribute to their genetic variation. We have tried to better explain this in the revised manuscript (lines 270-274).

Fig. c UPGMA clustering analysis of 27 *Chrysanthemum* accessions based on the Nei's genetic distance (Ding et al., 2007). Different *Chrysanthemum* genus species are colour and shape coded. Green stars represent the different geographical populations of *C. indicum*. Blue circles represent *C. morifolium*.

Fig. d Phylogenetic dendrogram of 26 groups based on RAPD data (Dai et al., 1998). The six *Chrysanthemum* genus species are colour coded. Artificial interspecific hybrids are indicated in gray font.

References:

- Wang, J., Yang, J., Li, M. (1993). The morphological variation and the karyotypical characters of *Dendranthema indicum* and *D. lavandulifolium*. *J. Syst. Evol.* 31, 140 (1993).
- Taniguchi, K. Cytogenetical studies on the speciation of tetraploid *Chrysanthemum indicum* L. with special reference to C-bands. *Jour. Sci., Hiroshima Univ.* 21, 105-157 (1987).
- Yang, W., Glover, B. J., Rao, G. Y., Yang, J. Molecular evidence for multiple polyploidization and lineage recombination in the *Chrysanthemum indicum* polyploid complex (Asteraceae). *New Phytol.* 171, 875-886 (2006).
- Ding, L., Chen, F., Fang, W. Genetic diversity among 27 materials in 8 species of *Dendranthema* by isozyme analysis. *Acta Bot. Bor-Occid. Sin.* 27, 249-256 (2007).
- Dai, S., Chen, J., Li, W. Application of RAPD analysis in the study on the origin of Chinese cultivated chrysanthemum. *Acta Botanica Sin.* 40, 1053-1059 (1998).

I was considering that *C. morifolium* was an auto-polyploid. However, the distinguished sequences among homoeologous chromosomes shown in this study suggested a possibility that *C. morifolium* is an allo-polyploid species. If it is allo-polyploid, the wild progenitor would not be a single species. Therefore, in the case the authors consider that *C. morifolium* was allo-polyploid species, I recommend making a phylogenetic tree in each chromosome, to confirm the number of possible progenitor species.

Response: This is indeed a very good suggestion of the reviewer that could help to further reveal the origin of chrysanthemum, which has always been a controversial issue. Here, we made phylogenetic trees for each of the 9 homoeologous groups based on the unique SNPs (depth > 20

& MAF > 0.05 & miss < 0.1). As a result, the three homoeologous chromosomes within a same group were all separated from each other except for Chr17 and Chr18. Notably, we found there was always a chromosome within one of the nine groups, respectively, Chr1, Chr4, Chr9, Chr10, Chr13, Chr16, Chr21, Chr22, Chr2 that was tightly clustered together with *C. rhombifolium*. This result was consistent with the phylogenetic analyses (Fig. 3a-c) suggesting that *C. rhombifolium* was more closely related to *C. morifolium*. In addition, there were one or two chromosomes that were tightly clustered with the other 8 *Chrysanthemum* wild species. For instance, Chr15 and Chr23 were both clustered with *C. indicum* (Hubei), while Chr5 and Chr11 were both clustered with *C. dichrum*. Chr19 was found tightly clustered with *C. nankingense*. This result suggests that diploid *C. nankingense* might not be a direct ancestral donor of *C. morifolium*, which may also help to answer the reviewer's next comment on gene retention analysis. We have added this analysis to the revised version of our manuscript (see Supplementary Note 5.1, Supplementary Fig. 17 and lines 303-307).

Fig. e Maximum-likelihood phylogenetic tree of *Chrysanthemum* for each of the nine homoeologous groups, using sunflower (*H. annuus*) as an outgroup.

L271-L274. It seems Supplementary Fig 14 does not show evidence of small introgression. How can we find the small integration from the figure?

Response: The ‘small introgression’ here indicates the small length of continuous windows with higher identical score (IS) values across the genome. We have replaced the ‘introgression’ with ‘hotspots’ to avoid confusion and re-wrote the sentence. Please see lines 275-280 in the revised manuscript.

L275-L284

I think the results suggested in Supplementary Fig 15 did not deny the possibility that *C.*

nankingense is a progenitor of *C. morifolium*, and rather suggest that

- the sequenced material was not fully auto-polyploid species
- *C. nankingense* would be a donor.

I recommend adding *C. seticuspe* genome reported by Nakano et al. (2021), as reference of the comparison between *C. nankingense* and *C. morifolium*.

Response: We agree with the reviewer's comment on the retention analysis that the sequenced material was not a fully auto-polyploid species. If so, there should not be significant differences among the three homoeologous chromosomes within a group. However, we could not make an absolute conclusion that *C. nankingense* is a direct donor of *C. morifolium* only from gene retention analysis, since there was no obvious chromosome pair exhibiting significantly higher gene retention ratios than any other two chromosomes. Moreover, the phylogenetic trees constructed in each group (Supplementary Fig. 17), mapping coverage depth (Supplementary Fig. 13), and IS analysis (Fig. 3c) suggest that *C. nankingense* might not be a direct donor of *C. morifolium*. In addition, by taking advantage of the high-quality *C. seticuspe* genome (Nakano et al., 2021), we made a comparison on the gene retention ratio between *C. nankingense* and *C. seticuspe*. As a result, 54.81% (40,702 out of 74,259) of the genes in *C. seticuspe* were syntenic with at least one subgenome in *C. morifolium*, among which 25,900 (~63.63%) syntenic genes that had three homoeologous copies retained in *C. morifolium* (Supplementary Fig. 16 and Supplementary Table 22). In general, the proportion of retained genes of *C. seticuspe* in *C. morifolium* was lower than that in *C. nankingense*, suggesting that *C. nankingense* might be more closely related with *C. morifolium* than with *C. seticuspe*. Please see lines 289-292 in the revised manuscript.

Table b Gene retention ratios between *C. nankingense* and *C. seticuspe* in the *C. morifolium* genome

Taxa	C. nankingense			C. seticuspe		
Group1	Chr1	Chr2	Chr3	Chr1	Chr2	Chr3
Mean	87.71	86.80	91.29	88.49	84.43	85.70
Sig.	ns	***	***	***	**	ns
Group2	Chr4	Chr5	Chr6	Chr4	Chr5	Chr6
Mean	89.52	89.48	82.97	86.03	86.51	80.77
Sig.	ns	***	***	ns	***	***
Group3	Chr7	Chr8	Chr9	Chr7	Chr8	Chr9
Mean	89.03	80.54	90.32	87.57	85.57	82.37
Sig.	***	ns	***	*	***	**
Group4	Chr10	Chr11	Chr12	Chr10	Chr11	Chr12
Mean	84.70	89.91	86.27	82.94	86.56	80.94
Sig.	***	ns	*	**	ns	***
Group5	Chr13	Chr14	Chr15	Chr13	Chr14	Chr15
Mean	90.98	87.77	83.33	89.63	85.43	81.16
Sig.	***	***	***	***	***	***
Group6	Chr16	Chr17	Chr18	Chr16	Chr17	Chr18

Mean	87.43	90.40	82.72	85.78	86.25	84.84
Sig.	***	***	***	ns	ns	ns
Group7	Chr19	Chr20	Chr21	Chr19	Chr20	Chr21
Mean	82.42	79.89	88.09	79.70	78.57	84.38
Sig.	ns	***	***	ns	***	***
Group8	Chr22	Chr23	Chr24	Chr22	Chr23	Chr24
Mean	86.24	89.04	85.86	85.52	86.33	84.66
Sig.	**	ns	**	ns	ns	ns
Group9	Chr25	Chr26	Chr27	Chr25	Chr26	Chr27
Mean	86.21	90.75	87.61	83.17	83.11	82.89
Sig.	***	ns	**	ns	ns	ns

Significance was estimated between Chr1 vs Chr 2, Chr1 vs Chr3, Chr2 vs Chr3, using a Student's *t* test (*** $P < 0.001$, ** $P < 0.01$, * $P < 0.05$, ns: not significance).

L305 I think Supplementary Fig. 17 rather suggested a possibility of AAB, not fully auto-polyploidy species.

Response: We agree with the reviewer and have adapted the text (lines 317-320 of the revised manuscript).

Chapter Homoeolog expression bias in hexaploid chrysanthemum

Supplementary Fig 19

I could not find the description about the organs extracted RNAs for the analysis in Supplementary Note 6. Please add details of the process in transcriptome sequencing. In addition, I could not understand how authors identify the chromosomes that express each gene, when homoeologous genes were located across chromosomes.

Response: The transcriptome datasets for different organs were obtained from our previous study (Ding et al., *Plant Molecular Biology*, 2020, 103: 669-688; NCBI BioProject ID: PRJNA548460) and we re-analyzed these data using the assembled *C. morifolium* genome as a reference. We have added a detailed description on sample information and transcriptome analysis in Supplementary Note 6 to clarify things for the reader. Besides, for accurate quantification of homoeologous gene expression, only the unique mapping reads were kept for further analysis (see Supplementary Note 6 of the revised paper).

Reference:

Ding, L. et al. The core regulatory networks and hub genes regulating flower development in *Chrysanthemum morifolium*. *Plant Mol. Biol.* 103, 669-688 (2020).

Supplementary Fig 20. What dose G1d~G3s mean?

Response: G1d~G3d and G1s~G3s represent the three dominant categories and three suppressed categories, respectively. Homoeolog expression bias analysis was performed according to a previous published method described for wheat (Ramírez-González et al., 2018). Briefly, we focused exclusively on the 11,438 expressed gene triads which have a 1:1:1 correspondence in syntenic blocks across the three homoeologous chromosomes. To standardize the relative

expression of each homoeolog across the triad, the absolute FPKM for each gene within the triad was normalized as follows (taken Chr1-Chr2-Chr3 for example):

$$\text{expression}_{\text{Chr1}} = \text{FPKM}_{\text{Chr1}} / (\text{FPKM}_{\text{Chr1}} + \text{FPKM}_{\text{Chr2}} + \text{FPKM}_{\text{Chr3}})$$

$$\text{expression}_{\text{Chr2}} = \text{FPKM}_{\text{Chr2}} / (\text{FPKM}_{\text{Chr1}} + \text{FPKM}_{\text{Chr2}} + \text{FPKM}_{\text{Chr3}})$$

$$\text{expression}_{\text{Chr3}} = \text{FPKM}_{\text{Chr3}} / (\text{FPKM}_{\text{Chr1}} + \text{FPKM}_{\text{Chr2}} + \text{FPKM}_{\text{Chr3}})$$

Subsequently, we defined seven homoeolog expression bias categories according to the relative expression of each homoeolog, as illustrated in Fig. f: a balanced category, with similar relative abundance of transcripts from the three homoeologs, and six homoeolog dominant (G1d/G2d/G3d) or homoeolog suppressed categories (G1s/G2s/G3s), with higher or lower abundance of transcripts from a single homoeolog with respect to those from the other two. We have expanded the method of homoeolog expression bias analysis in Supplementary Note 6 and explain the meaning of ‘G1d~G3s’ in the legend of this Supplementary Figure.

Fig. f Ternary plot (left) showing relative expression abundance of 1,568 syntenic triads (4,704 genes) in *C. morifolium* in the combined analysis of 14 tissues. Each circle represents a gene triad consisting of the relative contribution of each homoeolog to the overall triad expression. Triads in vertices correspond to single-chromosome-dominant categories, whereas triads close to edges and between vertices correspond to suppressed categories. Balanced triads are shown in gray. Box plots (right) indicate the relative contribution of each chromosome based on triad assignment to the seven categories. The fill color in the ternary plot was consistent with that in the box plots.

Reference:

Ramírez-González, R. H. et al. The transcriptional landscape of polyploid wheat. *Science* 361, eaar6089 (2018).

Reviewer #2 (Remarks to the Author):

In the presented article, titled, “A chromosome-scale genome assembly of hexaploid cultivated chrysanthemum”, Song et al., assembled a highly heterozygous and economically important ornamental plant, *Chrysanthemum morifolium* Ramat. Authors managed to assemble a difficult genome, and Hi-C contact map supported their claim of a nice genome resource. Authors identified a recent WGT event, and also suggested allopolyploid nature of *Chrysanthemum* genome with AAB composition. Authors have used strong evidence while adopting enough caution to claim WGT, allopolyploid nature of the genome, which is highly admirable. I find the

established resource from this study important step to explore and improve this valuable plant species, and was convinced with all the claims that authors made based on the described evidence. There are few minor issues that I hope authors could resolve mentioned below-

Response: Thanks for the very positive comments on our manuscript.

1. Line 108. In my understanding, Hi-C data orders the scaffold, but the sequencing data itself is not integrated. The sentence here is providing a different meaning. Please consider rephrasing it.

Response: We changed this sentence in our revised version of manuscript. Please see line 1.

2. Although Hi-C contact map does show 9 groups, each group with 3 elements, thus $n=27$, I wonder if this species has been used for karyotyping? If they could show the chromosome numbers, or provide a reference to support, that would be great. This will further validate their Hi-C data-based results.

Response: This is indeed a good suggestion. We successfully designed 12 oligonucleotide probes based on the putative satellites for FISH (fluorescence *in situ* hybridization) karyotyping in the sequenced diploid plant (see Supplementary Note 5.6 for detail). 11 of them resulted in visible signals covering the *C. morifolium* chromosomes (Fig. g). Surprisingly, most of the oligo-FISH karyotypes showed that two out of the three chromosomes within a group were generally more closely related than the third one except for the two groups with no signal. The Oligo-FISH assays also confirmed the difference of repeat sequences between the two similar “A” genomes and the existence of several chromosomal translocations in ‘Zhongshanzigui’ reference genome, which might support our conjecture about its “AA'B” genome structure. However, we were unable to precisely ascribe these karyotypes to the 27 anchored *C. morifolium* chromosomes due to the limitation of the used probes. More attempts need to be done in the future, such as design effective chromosome-specific probes. We have added a Supplementary Fig. 20 and discussed the results in lines 327-331 of the revised manuscript.

Fig. g RepeatExplore2-TAREAN analysis and Oligo-FISH results of the diploid plant of *C. morifolium* var ‘Zhongshanzigui’. **a** Graphic output of RepeatExplore2-TAREAN analysis based on Illumina reads. **b** Results of the 12 designed probes used for Oligo-FISH in *C. morifolium* (Bar=10 μ m). **c** The Oligo-FISH results by using CmOP-1(FAM, green) and CmOP-2(TAMRA, red) (Bar=10 μ m). **I** DAPI channel, **II** green channel, **III** red channel, **IV** merge channel, **V** the Oligo-FISH karyotype. White dotted frame showed the two signal-less chromosome groups. **VI** The chromosome idiograms of *C. morifolium* based on Oligo-FISH results. The signals of CmOP-1 (green) and CmOP-2 (red) were marked in circle (Bar = 10 μ m).

3. Line 114, “clear cut” sounds typo or strange. Please consider rephrasing it.

Response: Thanks. We have changed it in lines 113-114.

4. Please provide number of gaps within this genome in the manuscript text. It will provide a sense of completeness and quality of genome. Given the heterozygosity and the repeat content of this plant genome, I feel that the contig N50 is quite decent.

Response: Thanks for your appreciation of the genome assembly of *C. morifolium*, a polyploid species with a large repeat-rich genome. The total number of gaps in *C. morifolium* genome was 13,556, ranging from 317 (Chr5) to 814 (Chr18). Gap sequences account for 0.36% of the total genome sequences (Supplementary Table 4). We have added the information on gap numbers in Supplementary Table 4.

5. Authors should consider comparing individual groups (9 groups that they identified) based on synteny, and should try and see if the gene order is changed, and if that has any implication for flower color or shape or other biological processes. Authors did mentioned differences in the expression of genes within groups (for example between chr13 and chr14), but what percentage of synteny they observed within each groups? Is this same or similar across all groups? Does order of genes remains the same? What changes whatsoever they could observed?

Response: We analyzed synteny in each of the nine homoeologous chromosome groups and observed a greater difference of synteny in inter-group (25.3% to 40.8%) than in intra-group. About the orientation of the synteny genes in gene locus, we found 79.7% to 94.6% of them to be the same (see Supplementary Table 25 for detail). Further, we analyzed the relationship between gene orientation and gene expression and found a weak correlation between those (Supplementary Table 26), suggesting that homoeolog expression bias might not be caused by the change of gene orientation. We have added these results to the revised manuscript (lines 354-357 and Supplementary Note 6).

Table c Correlation between gene order change and gene expression change

Group	Correlation coefficient;	P value
Chr1-Chr2-Chr3	-0.0821	1.14E-03
Chr4-Chr5-Chr6	-0.0174	5.77E-01
Chr7-Chr8-Chr9	-0.0514	6.48E-02
Chr10-Chr11-Chr12	-0.0417	1.41E-01
Chr13-Chr14-Chr15	-0.0430	1.10E-01
Chr16-Chr17-Chr18	0.0554	8.42E-02

Chr19-Chr20-Chr21	-0.0201	5.50E-01
Chr22-Chr23-Chr24	0.0204	4.05E-01
Chr25-Chr26-Chr27	0.0265	3.21E-01
All	-0.0189	4.34E-02

6. Authors used Wild type species to ascertain ploidy nature of Chrysanthemum, and reported 104597 synonymous SNPs. I am curious if these SNPs had any associated with features that are involved with the coloring or shape of flowers. Compared to these wild types, is their any specific genomic regions that showed over representation of these SNPs? Any biological interpretations that one could derive from it?

Response: This is indeed an interesting question. Here, we counted the number of high-quality SNPs (depth > 20 & MAF > 0.05 & miss < 0.1) in 1 Mb non-overlapping sliding windows. The genes within the top 5% hot spots of SNP distribution regions were selected and used to perform GO enrichments analysis. The results showed that the ‘auxin homeostasis’ and ‘regulation of flavonoid biosynthetic process’ were the most enriched biological processes terms that might respectively involve floral development and coloring (Supplementary Table 30). We have added the descriptions in Supplementary Note 5.1 of the revised manuscript.

7. Line 238, sentence seems very strange, some mistake. Please check and correct it.

Response: Corrected. Please see lines 238-239 of the revised manuscript.

8. Please provide your interpretation on different copia-to-gypsy ration that they observed in discussion section.

Response: We compared the *Copia*-to-*Gypsy* ratio in the Asteraceae species whose genomes have been completely sequenced and published and re-organized the Results section in lines 240-246.

9. The color of flowers certainly involved several secondary metabolic pathways including phenylpropanoid among others. Authors should consider describing this aspect using the generated dataset. This will improve scope of this manuscript to a broader audience.

Response: Thanks for this suggestion. Previous studies have characterized individual genes encoding relevant enzymes underlying pigment formation for floral crops. Based on the chromosome-scale genome sequence of *C. morifolium* presented here, we selected three Santini series chrysanthemum cultivars (‘Mini Pink’, ‘Mini Yellow’ and ‘Mini White’) with similar genetic background but different flower colors to investigate the biosynthesis pathway of anthocyanin and flavonol by conducting an RNA-seq analysis. The results showed that the expression levels of anthocyanin biosynthesis genes in ‘Mini Pink’ were generally higher than that of ‘Mini Yellow’ and ‘Mini White’. Nevertheless, the expression level of *CmFLS* was distinct in ‘Mini Yellow’, suggesting that the highly expressed *CmFLS* in ‘Mini Yellow’ competed with the same substrate of the flavonol pathway and anthocyanin pathway, thus resulting in more colorless flavonols, which contributed to its lack of pink. Since there were no *CCD4a* genes annotated in the ‘Zhongshanzigui’ reference genome, we further made a *de novo* transcriptome assembly and found that the expression of *CmCCD4a* in ‘Mini Yellow’ was significantly lower than that of ‘Mini Pink’ and ‘Mini White’, which was consistent with the carotenoid content in the three cultivars. Collectively, our results indicate that the flower color diversity is a combined

consequence of anthocyanin and carotenoid metabolic pathways in chrysanthemum. We have added the main conclusion in lines 425, 496-498, and provided details in Supplementary Note 8 and Supplementary Fig. 30.

Fig. h The biosynthesis pathways of anthocyanin and flavonol. **a** Photograph shows the capitulum phenotypes of ‘Mini Pink’ (Mini P), ‘Mini Yellow’ (Mini Y) and ‘Mini White’ (Mini W). Bar=1 cm. **b** Bar graph shows the relative total content of anthocyanins in the ray floret petals of three Santini chrysanthemum cultivars. **c** Bar graph shows the relative content of carotenoids in the ray floret petals of three Santini chrysanthemum cultivars. Error bars indicate the standard deviation (*SD*) of three biological replicates. Significant differences are indicated with asterisks (***) $P < 0.001$, ANOVA, Tukey’s correction). **d** Gene expression profile of biosynthesis genes in flavonoids biosynthesis pathway of petals among three cultivars (from left to right in each heatmap panel are Mini P, Mini Y, Mini W, respectively, with three biological replicates) are

presented in the heatmap alongside the gene names. The bar represents the expression level of each gene (z-score). Low to high expression is indicated by a change in color from blue to red. e The FPKM value of Unigene0040398 (*CmCCD4a*) with three biological replicates among three cultivars.

Overall, the genome quality and the interpretation for this manuscript is sound and impressive. The genomic resource itself is valuable, and therefore, with these changes, I find this manuscript an important advancement for future studies on Chrysanthemum including species improvement and novel biology based discoveries.

Response: Thanks again for the positive comments on our manuscript. We have improved the manuscript according to the Reviewers' suggestions. We believe the release of cultivated chrysanthemum reference genome would give further insights into the evolutionary history of chrysanthemum and will facilitate the identification of genes that are related to important horticultural traits through forward genetics approaches. This genome might also provide a valuable reference for other complex polyploids that are not of strict allo- or auto-polyploid origin.

Reviewer #3 (Remarks to the Author):

Chrysanthemum is one of the most widely cultivated ornamental plants, with great economic and cultural values. For a long time, the various ploidy has limited the research for chrysanthemum species, especially under their complex evolutionary history (e.g., the frequent interspecific hybridizations). Due to the limitation of sequencing and assembling technologies, it is really difficult to generate the high-quality genome assembly for the polyploid species. Even now, the reported genomes of hexaploidy species are quite rare. In this manuscript, Song et al. reported a newly assembled genome of hexaploid cultivated chrysanthemum, *Chrysanthemum morifolium* Ramat., explored its evolutionary history (including the genome structure, WGD and WGT events, and origin history), and further identified the novel genes controlling key ornamental traits with multiple evidences. In summary, this is an exciting work, which provides an extremely valuable genetic resource for researchers in the field and could facilitate the related research (e.g., molecular breeding) for chrysanthemum species.

Response: Thanks for the positive comments on our manuscript.

Overall, I am glad to review this nice work. I am pleased to recommend it for publication after the revision according to my following major and minor comments.

Major:

1. To study the origin of *C. morifolium*, the authors mapped the reads of 12 wild species to *C. morifolium* genome and further reconstructed the phylogenetic topology using their SNPs. However, the ploidy of sampled species is various, including diploid, tetraploid, and hexaploid. For phylogenetic analysis, it is unreasonable to map the samples with such various ploidy to a single reference genome and directly generate the phylogenetic tree based on such obtained SNPs. So, the deduced origin of *C. morifolium* is not credible. Actually, the common way to study the origin of polyploid species (tetraploid and/or hexaploidy species) is to divide their genomes into different subgenomes and further construct their phylogenetic relationships. Furthermore, the

comprehensiveness of sampling will directly affect the result. The authors should declare whether all wild relatives are sampled in this study. It is better to provide a map of their distribution areas in the manuscript.

Response: We agree with the reviewer's comment that the common way to study the origin of polyploid species is to divide their genomes into different subgenomes. However, unlike the most frequently mentioned polyploid species such as wheat, cotton, peanut, *Brassica*, alfalfa, sugarcane, strawberry, etc., there was no clear historical record or any study that could suggest a certain direct donor of cultivated chrysanthemum. *C. nankingense* is considered as a potential ancestor of cultivated chrysanthemum (Song et al., 2018). Here, we generated a high-quality chromosome-level genome assembly of *C. nankingense*. But we failed to utilize it to distinguish the subgenomes, which might be due to *C. nankingense* not being a direct ancestor of cultivated chrysanthemum. China is recognized as the country of origin of cultivated chrysanthemum and the lower-middle reaches of the Yangtze River areas including Chongqing, Hubei, Jiangsu, Anhui as well as Henan were the most likely the centers of origin (Chen et al., 2012). Several wild *Chrysanthemum* species are speculated to have contributed to the present complex chrysanthemum cultivars (Chen et al., 2012). Hence, we selected almost all extant Chinese diploid wild *Chrysanthemum* species as well as the *C. indicum* with multiple ploidies that have been reported to be the potential ancestors of cultivated chrysanthemum to investigate the origin of chrysanthemum. The distribution of these wild species is shown in the Fig. i. We have provided their locations in Supplementary Table 19 and highlighted the description of sampling comprehensiveness in lines 258-260, 598-599 and Supplementary Note 5.1. Our analysis was based on the published method in Nature Genetics (Zhang et al., 2018), where the population genetic structure and phylogenetic relationships among 64 *S. spontaneum* accessions from hexaploid to hexadecaploid were analyzed by using the assembly haploid genome of octoploid *S. spontaneum* as a reference.

Fig. i Distribution within China of the 12 resequenced wild *Chrysanthemum* species in this study

References:

Chen, J., Wang, C., Zhao, H. & Zhou, J. The origin of garden Chrysanthemum. Anhui Science & Technology Publishing House, Hefei (2012).

Song, C. et al. The *Chrysanthemum nankingense* genome provides insights into the evolution and diversification of chrysanthemum flowers and medicinal traits. *Mol. Plant* 11, 1482-1491 (2018).

Zhang, J. et al. Allele-defined genome of the autopolyploid sugarcane *Saccharum spontaneum* L. *Nat. Genet.* 50, 1565-1573 (2018).

2. The authors argued that *C. morifolium* is an allopolyploid with the genome structure of “AAB” via multiple evidences. However, when mapping to the reference genome, a pattern of uniform coverage of different wild species was observed. It is strange and unreasonable. If *C. morifolium* is an allopolyploid (“AAB”), when mapping diploid or tetraploid species to it, we could expect an obvious different coverage depth between A and B subgenomes. Otherwise, if the methods for read mapping applied correctly, it suggests *C. morifolium* is more likely to be an autopolyploidy (“AAA”). The authors should justify these conflicts.

Response: Here, we tried our best to clarify the origin of chrysanthemum, which has always been a controversial issue. According to evidence from genome architecture, molecular markers and cytological experiments provided in this study, together with evidence from previous reports, as well as the comparisons with other typical allopolyploids and autopolyploids, we suggest the sequenced chrysanthemum material is likely to be a ‘segmental allopolyploid’ (Stebbins Jr, 1947), i.e., neither a strict allopolyploid nor a strict autopolyploid, in both the first and the revised version of our manuscript. Nevertheless, the added SNP- (Fig. e) or single copy gene- (Fig. a) based phylogenetic analysis as well as the FISH karyotypes (Fig. g) supported the sequence difference among the homoeologous chromosomes and the ongoing divergent evolution of *C. morifolium*. Hence, we would prefer to speculate that the sequenced material is more likely to be “AA'B” rather than “AAB” or “AAA”. Please also see our detailed responses to the third major comment of Reviewer 1 and the second comment of Reviewer 2. Besides, we do understand the reviewer’s concern about the observed uniform coverage depth among different wild species. We speculate that the results might be the consequence of two things: (1) Due to the close relationship and natural interspecific hybridization characteristics of the *Chrysanthemum* genus, the extant diploid or tetraploid wild *Chrysanthemum* species have also undergone extensive reticulate evolution with interspecific hybridization, introgression and polyploidization; (2) The most primitive ancestor of cultivated chrysanthemum is predicted to be probably extinct, which was also reported by Ma et al. (2020). We explain this in lines 310-314 of the revised manuscript and further clarified the “Origin of cultivated chrysanthemum” section.

References:

Stebbins Jr, G.L. Types of polyploids: their classification and significance. *Adv. Genet.* 1, 403-429 (1947).

Ma, Y.P. et al. Origins of cultivars of Chrysanthemum—Evidence from the chloroplast genome and nuclear LFY gene. *J. Syst. Evol.* 58, 925-944 (2020).

3. The authors amplified different alleles of CCD4a and speculated they might contribute to the

flower color diversity of modern cultivated chrysanthemums. It is understandable that, for such hexaploidy non-model plant species, it is hard to perform the in vivo functional verification tests. However, more evidence (e.g., data from the transcriptomes) should be provided to confirm such inference. Otherwise, it needs turn down the conclusion.

Response: Reviewer 2 also described the same problem. Here, we re-conducted an RNA-seq analysis using three Santini series chrysanthemum cultivars that had a similar genetic background but different flower colors to investigate both the expression patterns of genes encoding key biosynthetic enzymes in color determination and the role of *CCD4a* in flower color. Please see our detailed response to the ninth comment of Reviewer 2. We have added the main conclusion in lines 423, 504-506 and provided details in Supplementary Note 8 and Supplementary Fig. 30 of the revised manuscript.

Minor:

1. L35: Change “Million years ago” to “million years ago”.

Response: Changed.

2. L128: Please provide the full Latin name of *E. chinochloa* when it occurs for the first time.

Response: Revised

3. L177: Change “Functional annotation” to “Functional enrichment analysis”.

Response: Changed

4. L192-L210: The naming system of WGD/WGT (“WGT-1”, “WGT- γ ”, “WGD-2”, and “WGT-2” et al.) is really hard to follow. I suggest the authors to label them in Fig. 2a on corresponding positions at the same time. Moreover, why there is no “WGD-1”?

Response: We have added the labels of WGD/WGT events in Fig. 2a. To avoid confusion, the nomenclature of asterids II shared WGT-1 event and sunflower-specific WGD-2 event was referred to the published article of sunflower genome. However, there was no description about “WGD-1” in Badouin et al. (2017). We guess the ‘1’ and ‘2’ were used in chronological order leaving “WGT” or “WGD” out of the consideration.

Reference:

Badouin, H. et al. The sunflower genome provides insights into oil metabolism, flowering and Asterid evolution. *Nature* 546, 148-152 (2017).

5. L258: Whether the 12 samples used in this study are different species. I notice some of them look like different populations/ecotypes/varieties. The statistics of 12 samples should be provided in Supplementary Tables, e.g., their sampled locations, sequencing data size, mapping coverage, and mapping depth et al.

Response: The selected 12 *Chrysanthemum* materials used for re-sequencing were from 6 species, among which the *C. indicum* plants that were assigned to seven geographical populations. We have added the statistics of the 12 samples in Supplementary Table 19.

6. L263: According to the Supplementary Note, the SNPs used for phylogenetic analysis are 4DTV sites. If so, please change “synonymous” to “4DTV”. More details should be provided in the Methods or Supplementary Notes. For example, the authors should declare why the analysis was based on 4DTV sites, but not the genome-wide SNPs or intergenic SNPs.

Response: We apologize for the wrong description in the methods of phylogenetic analysis. The original ML tree and the updated ML tree (depth > 20 & MAF > 0.05 & miss < 0.1) were constructed both using the genome-wide SNPs to utilize the full variation information. We have revised the relevant descriptions in the manuscript and Supplementary Note 5.1.

7. L264-L265: As I mentioned in the Major Comment 1, I do not think the phylogenetic relationships constructed via such way is believable. In addition, based on this phylogenetic tree (Fig. 3a), *C. dichrum* and *C. indicum* (Hubei) may be the same closely to *C. morifolium* as *C. indicum* (Nanjing).

Response: We have updated the phylogenetic tree using a SNP set that was obtained with a more stricter SNP filter criteria (depth > 20 & MAF > 0.05 & miss < 0.1), as to Reviewer 1’s suggestion. From the results of phylogenetic and IS values (Please see the next response), we found that *C. rhombifolium*, *C. indicum* (Nanjing), *C. dichrum*, *C. indicum* (Hubei), *C. potentilloides* and *C. indicum* (Tianzhushan) were more closely to *C. morifolium*. Please see revised manuscript, lines 262-265.

8. L268-L272: Why the rank of identity score (IS) is not consistent to the phylogenetic tree (Fig. 3a). For example, *C. rhombifolium* is the most closely to *C. morifolium* in Fig. 3a, but the IS of *C. rhombifolium* (0.832) is less than that of *C. indicum* from Nanjing (0.859), Tianzhushan (0.840) and Hubei (0.837), and *C. dichrum* (0.834). I am sorry to say it again, does it also reflect the method applied for phylogenetic analysis is unreasonable, as I mentioned in Major Comment 1. Please justify it.

Response: We apologize for the mistake. The order of IS values from high to low are *C. indicum* (Nanjing) (0.866), *C. rhombifolium* (0.853), *C. indicum* (Tianzhushan) (0.847), *C. indicum* (Hubei) (0.844), *C. dichrum* (0.841), *C. potentilloides* (0.833), *C. nankingense* (0.816), *C. lavandulifolium* (0.810), *C. indicum* (Wuyishan) (0.799), *C. indicum* (Yuntaishan) (0.782), *C. indicum* (Shennongjia) (0.777), *C. indicum* (Henan) (0.768). We made a mistake in describing the results of IS analysis in the first version of manuscript. Three light blue horizontal lines were drawn in the original Fig. 3c to compare the IS values. We also provided the source data in Source Data file 1. Please kindly check it. Generally, the rank of identity scores was consistent with the updated phylogenetic tree (Fig. 3a). The top 2 highest IS value species *C. indicum* (Nanjing) and *C. rhombifolium* were tightly clustered with *C. morifolium*. The four species with the lowest IS values, i.e. *C. indicum* (Wuyishan), *C. indicum* (Yuntaishan), *C. indicum* (Shennongjia), *C. indicum* (Henan), were also clustered together and separated from *C. morifolium*, *C. indicum* (Tianzhushan), *C. indicum* (Hubei), *C. dichrum* and *C. potentilloides* with intermediate IS values were clustered between the two branches. Please see lines 265-270 of the revised manuscript.

Fig. j The original Fig. 3c adding three light blue horizontal lines to compare the IS values. We also added the IS values and significance of difference (means with different letters indicate significantly different at $P < 0.01$) on the top of the box plots in the revised version.

9. Methods & Supplementary Notes: More details should be provided in the Methods and/or Supplementary Notes. There is so less information on how the authors performed the analyses. The readers could not know the details of methods and thus could not further evaluate the accuracy of some results. Please improve the sections of Methods & Supplementary Notes.

Response: We have added details to the Methods & Supplementary Notes in the revised version.

10. L561-L569: The list of query genes and the information of the domains should be provided.

Response: We have deposited the list of query genes to figshare (https://figshare.com/articles/book/The_list_of_query_genes/21610305) and added the information of the domains in Supplementary Note 3.3.

11. L599: I believe the editor would confirm all data has been released before publication.

Response: We have submitted all related data to NCBI PRJNA796762, PRJNA895586 and figshare (https://figshare.com/articles/book/The_list_of_query_genes/21610305). We will release them immediately once the manuscript is accepted. The source data underlying Figs. 1a, 1c, 2a-d, 3b-d, as well as Supplementary Figs. 7b, 19, 22, 23, 24 are provided in Source Data file1.

12. Fig. 2a. All fossil calibrations used in the study should be listed and cited.

Response: The four fossil calibration points used in this study were obtained from TIMETREE, a public knowledge-base for information on the evolutionary timescale of life. The provided median and adjusted times between two species in TIMETREE was derived from many published studies. We have added the website address of TIMETREE and its citation in the legend of Fig. 2.

13. Fig. 3a: Please confirm whether the phylogenetic tree is constructed by ML methods. TreeBest is always applied in phylogenetic analyses based on NJ methods. The ploidy labeled on the right is too small.

Response: The phylogenetic tree was indeed constructed by ML methods instead of TreeBest. Here, the ML tree was re-constructed using IQ-TREE v1.6.12 (-bb 1000) (Nguyen et al., 2015) based on a more strictly screened SNP sets (depth > 20 & MAF > 0.05 & miss < 0.1) as per Reviewer 1's suggestion. We have revised the Methods & Note. The ploidy labels were enlarged and moved to below the sample names.

Reference:

Nguyen, L. T., Schmidt, H. A., Von Haeseler, A. & Minh, B. Q. IQ-TREE: a fast and effective stochastic algorithm for estimating maximum-likelihood phylogenies. *Mol. Biol. Evol.* 32, 268-274 (2015).

14. Fig. 3c & Supplementary Fig. 8: It is difficult to observe the differences between the average values of different groups. The statistical test could be performed, or the figure could be improved.

Response: We have performed the statistical test and added the significant difference results in Fig. 3c and Supplementary Fig. 8.

15. Fig. 4c: Please annotate and change the gene ID on the right to the gene names.

Response: We have added the gene names in Fig. 4c.

16. Supplementary Fig. 5: The unit of the P value in the figure (the heat map) should be labeled.

Response: Revised it.

17. Supplementary Fig. 9: Typo? Should the unit of P-adjust value be Log10 or?

Response: We apologize for the typo. It should be '-Log10'. We have revised it.

18. Table 1: Typo. Change "BUSCO completeness of annoation" to "BUSCO completeness of annotation".

Response: Corrected.

Reviewers' Comments:

Reviewer #1:

Remarks to the Author:

The manuscript is well revised from the original. It is especially impressive that the authors added the FISH analysis data. The discussion about genome structure is also more in-depth.

In reading the revised manuscript, a few new questions arose. I believe that a revised manuscript taking the following points taking in account would make the manuscript even more valuable.

L130 and Table S8

What is 'homology SNPs'. I've never heard the phrase. Dose it means 'homozygous' SNPs?

L171-L172: Here, we selected the longest chromosome within each homoeologous group to represent monoploid chrysanthemum in this analysis.

Please snow which chromosomes were selected in the main text.

In the previous review, I considered that *C. morifolium* was auto-polyploid species, and did not care so much the approach, that is selecting the longest chromosome in homoeologous group. However, with the response from the author, I understand that *C. morifolium* is rather allo-polyploid species.

In addition, the results shown in Supplementary Fig, 15 showed that 76% syntenic genes had three homoeologous copies retained in *C. morifolium*. It means, 24% genes would be chromosome specific. With the consideration of gene sequence differences, I wonder whether the approach (selecting the longest chromosome for creating monoploid *C. morifolium*) adequate for the comparative genomics and evolutionary analysis.

Should authors create a consensus gene set and use it for the analysis instead of choosing the longest chromosome?

L202-204: These results provide additional evidence for the inferred recent polyploidy event in the common ancestor of *Chrysanthemum* species to be more likely a WGT event, rather than WGD (here termed WGT-2).

We see 3:1 syntenic relationship within *C. morifolium* monoploid pseudochromosomes and between *C. morifolium* and globe artichoke. However, those synteny were partial, not entire genome. It seems that the triplication was rather happened partially, not 'Whole genome'. How do you think?

Reviewer #2:

Remarks to the Author:

In the presented revision of the manuscript, titled, "A chromosome-scale genome assembly of cultivated chrysanthemum", authors have addressed all of my concern, and have added wealth of new data, which is very impressive. I am very happy with the additional value-added data that significantly improves the article, and now support its publication. I do advice authors to consider adjusting the title of this article as this sounds to general to me, but i guess the journal and authors will have a chance to modify the title to have more impactful message that may come across through the title.

Reviewer #3:

Remarks to the Author:

Most of my previous concerns have been addressed in the revised edition. I have only two small points that need to be revised.

1. The total ms is better to be sent to one native English speaker for being polished. Although most contents could not be understood, English needs to be improved.

2. About segmental allopolyploid, I suggest to just use 'allopolyploid'. Both AA genomes may have derived from two very similar species without distinct differentiation. If this is the case, both AA genomes could be treated as a tetraploid. In addition, species concept and specie delimitation in this genus are unclear. I suggest to delete the discussion related to 'strict autopolyploid or allopolyploid'.

Detailed Response to Reviewers

Reviewer #1 (Remarks to the Author):

The manuscript is well revised from the original. It is especially impressive that the authors added the FISH analysis data. The discussion about genome structure is also more in-depth.

Response: We want to thank the reviewer for his/her positive comments on our revised manuscript.

In reading the revised manuscript, a few new questions arose. I believe that a revised manuscript taking the following points taking in account would make the manuscript even more valuable.

L130 and Table S8

What is 'homology SNPs'. I've never heard the phase. Dose it means 'homozygous' SNPs?

Response: Yes. We have replaced "homology" by "homozygous" in line 130 and Table S8 of the revised manuscript.

L171-L172: Here, we selected the longest chromosome within each homoeologous group to represent monoploid chrysanthemum in this analysis.

Please snow which chromosomes were selected in the main text.

In the previous review, I considered that *C. morifolium* was auto-polyploid species, and did not care so much the approach, that is selecting the longest chromosome in homoeologous group. However, with the response from the author, I understand that *C. morifolium* is rather allo-polyploid species. In addition, the results shown in Supplementary Fig, 15 showed that 76% syntenic genes had three homoeologous copies retained in *C. morifolium*. It means, 24% genes would be chromosome specific.

With the consideration of gene sequence differences, I wonder whether the approach (selecting the longest chromosome for creating monoploid *C. morifolium*) adequate for the comparative genomics and evolutionary analysis.

Should authors create a consensus gene set and use it for the analysis instead of choosing the longest chromosome?

Response: We agree with the reviewer and have now created a consensus gene set for comparative genomics analysis. The idea of creating a consensus gene set is to remove redundant genes within a homoeologous group while including as much genetic information as possible. The longest genes among each set of orthologous gene pairs, together with the chromosome-unique genes constituted the final consensus gene set amounting to 94,552 genes in total. Among these homoeologous groups, 14.29% (13,507/94,552) and 13.63% (12,884/94,552) have three and two homoeologous genes, respectively. The remaining 68,161 (72.08%) genes are chromosome-unique genes.

Based on the consensus gene set, we obtained the same phylogenetic tree topology and similar divergence times, as compared with the previous analysis (Fig. a). Notably, the enriched GO terms of the expanded gene families tend to exhibit increased biological significance (Supplementary Fig 5 and Table 17), indicating the suitability of the new consensus gene set. We have revised the "Comparative genomics and evolutionary analysis" section (see lines 171-188 and 574-580) and re-organized Fig.2a, Supplementary Note 4.1, Supplementary Figs 3-5 and Supplementary Tables 16 and 17.

Fig. a Species phylogenetic trees obtained by RAxML using the new consensus gene set (top; 491 single-copy gene families), and previous gene set (bottom; 538 single-copy gene families) for *C. morifolium*.

L202-204: These results provide additional evidence for the inferred recent polyploidy event in the common ancestor of *Chrysanthemum* species to be more likely a WGT event, rather than WGD (here termed WGT-2).

We see 3:1 syntenic relationship within *C. morifolium* monoploid pseudochromosomes and between *C. morifolium* and globe artichoke. However, those synteny were partial, not entire genome. It seems that the triplication was rather happened partially, not 'Whole genome'. How do you think?

Response: This is indeed an interesting question. Actually, we found many chromosome segments showing 3 to 1 syntenic relationships within both the *C. morifolium* monoploid genome and *C. nankingense* genome (Fig. b), not limited to the marked information in Supplementary Fig. 6. The

high proportion as well as the extensive and random distribution of the syntenic relationships (i.e., the synteny was not limited to one or few chromosomes), suggesting the less possibility of segmental duplications. It is a common situation that massive gene loss or even chromosomal rearrangement occurred after the WGD or WGT events in plant evolutionary history (Sankoff & Zheng, 2018). Therefore, we tend to think the observed partial but high proportion of the 3 to 1 syntenic relationships might due to the tachytelic evolution of *Chrysanthemum* species, as we mentioned in the manuscript.

Fig. b Dot plots showing the syntenic relationships within *C. morifolium* monoploid genome (left) and *C. nankingense* genome (right). The 3 to 1 syntenic relationships are alternately shaded in gray and pink.

Reference:

Sankoff, D., & Zheng, C. Whole genome duplication in plants: implications for evolutionary analysis. In *Comparative Genomics* (pp. 291-315). Humana Press, New York, NY (2018).

Reviewer #2 (Remarks to the Author):

In the presented revision of the manuscript, titled, "A chromosome-scale genome assembly of cultivated chrysanthemum", authors have addressed all of my concern, and have added wealth of new data, which is very impressive. I am very happy with the additional value-added data that significantly improves the article, and now support its publication. I do advice authors to consider adjusting the title of this article as this sounds to general to me, but i guess the journal and authors will have a chance to modify the title to have more impactful message that may come across through the title.

Response: Thanks for the constructive suggestion. We have changed our manuscript title as "A chromosome-scale genome assembly reveals the origin and evolution of cultivated chrysanthemum" to reflect more impactful message.

Reviewer #3 (Remarks to the Author):

Most of my previous concerns have been addressed in the revised edition. I have only two small points that need to be revised.

1. The total ms is better to be sent to one native English speaker for being polished. Although most contents could not be understood, English needs to be improved.

Response: We have sent this revised version of manuscript to the AJE company for English polish. We also asked Prof. Yves Van de Peer to review our manuscript again, with special attention to the English grammar and wording. Please see the revised manuscript for details.

2. About segmental allopolyploid, I suggest to just use 'allopolyploid'. Both AA genomes may have derived from two very similar species without distinct differentiation. If this is the case, both AA genomes could be treated as a tetraploid. In addition, species concept and specie delimitation in this genus are unclear. I suggest to delete the discussion related to 'strict autopolyploid or allopolyploid'.

Response: We agree with the reviewer that the AA' genomes might have derived from two very similar wild species without distinct differentiation. However, according to the multi-level evidence provided in this study, we would still prefer to speculate that the sequenced material 'Zhongshanzigui' is a segmental allopolyploid (as defined by Stebbins et al., (1947)), rather than a full allopolyploid. Nevertheless, considering the concepts of both forms of polyploid and the *Chrysanthemum* genus are ambiguous, we agree with the reviewer and have deleted the discussion related to "strict autopolyploid or allopolyploid" as suggested (see lines 334-346).

Reference:

Stebbins Jr, G.L. Types of polyploids: their classification and significance. *Adv. Genet.* **1**, 403-429 (1947).

Reviewers' Comments:

Reviewer #1:

Remarks to the Author:

The manuscript is well revised from the previous version. Now i recommend its acceptance.

Reviewer #3:

Remarks to the Author:

I have no furthr conern.

Detailed Response to Reviewers

Reviewer #1 (Remarks to the Author):

The manuscript is well revised from the previous version. Now i recommend its acceptance.

Response: Thanks for the positive decision on our manuscript.

Reviewer #3 (Remarks to the Author):

I have no further concern.

Response: Thanks.